# Noisy Feature Mixup

**Soon Hoe Lim**[*]
Nordita,
KTH and Stockholm University
soon.hoe.lim@su.edu

**N. Benjamin Erichson**[*]
University of Pittsburgh
erichson@pitt.edu

**Francisco Utrera**
University of Pittsburgh
and ICSI
utrerf@berkeley.edu

**Winnie Xu**
University of Toronto
winniexu@cs.toronto.edu

**Michael W. Mahoney**
ICSI and UC Berkeley
mmahoney@stat.berkeley.edu

## Abstract

We introduce Noisy Feature Mixup (NFM), an inexpensive yet effective method for data augmentation that combines the best of interpolation based training and noise injection schemes. Rather than training with convex combinations of pairs of examples and their labels, we use noise-perturbed convex combinations of pairs of data points in both input and feature space. This method includes mixup and manifold mixup as special cases, but it has additional advantages, including better smoothing of decision boundaries and enabling improved model robustness. We provide theory to understand this as well as the implicit regularization effects of NFM. Our theory is supported by empirical results, demonstrating the advantage of NFM, as compared to mixup and manifold mixup. We show that residual networks and vision transformers trained with NFM have favorable trade-offs between predictive accuracy on clean data and robustness with respect to various types of data perturbation across a range of computer vision benchmark datasets.

## 1 Introduction

Mitigating over-fitting and improving generalization on test data are central goals in machine learning. One approach to accomplish this is regularization, which can be either data-agnostic or data-dependent (e.g., explicitly requiring the use of domain knowledge or data). Noise injection is a typical example of data-agnostic regularization (Bishop, 1995), where noise can be injected into the input data (An, 1996), or the activation functions (Gulcehre et al., 2016), or the hidden layers of deep neural networks (Camuto et al., 2020; Lim et al., 2021). Data augmentation constitutes a different class of regularization methods (Baird, 1992; Chapelle et al., 2001; DeCoste & Schölkopf, 2002), which can also be either data-agnostic or data-dependent. Data augmentation involves training a model with not just the original data, but also with additional data that is properly transformed, and it has led to state-of-the-art results in image recognition (Cireşan et al., 2010; Krizhevsky et al., 2012). The recently-proposed data-agnostic method, mixup (Zhang et al., 2017), trains a model on linear interpolations of a random pair of examples and their corresponding labels, thereby encouraging the model to behave linearly in-between training examples. Both noise injection and mixup have been shown to impose smoothness and increase model robustness to data perturbations (Zhang et al., 2020; Carratino et al., 2020; Lim et al., 2021), which is critical for many safety and sensitive applications (Goodfellow et al., 2018; Madry et al., 2017).

In this paper, we propose and study a simple yet effective data augmentation method, which we call *Noisy Feature Mixup* (NFM). This method combines mixup and noise injection, thereby inheriting the benefits of both methods, and it can be seen as a generalization of input mixup (Zhang et al., 2017) and manifold mixup (Verma et al., 2019). When compared to noise injection and mixup, NFM imposes regularization on the largest natural region surrounding the dataset (see Fig. 1), which may help improve robustness and generalization when predicting on out of distribution data. Conveniently, NFM can be implemented on top of manifold mixup, introducing minimal computation overhead.

---

[*]Equal contribution

**Contributions.**    Our main contributions are as follows.

- We study NFM via the lens of implicit regularization, showing that NFM amplifies the regularizing effects of manifold mixup and noise injection, implicitly reducing the feature-output Jacobians and Hessians according to the mixing level and noise levels (see Theorem 1).

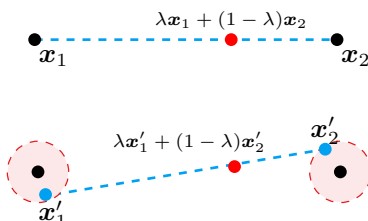

- We provide mathematical analysis to show that NFM can improve model robustness when compared to manifold mixup and noise injection. In particular, we show that, under appropriate assumptions, NFM training approximately minimizes an upper bound on the sum of an adversarial loss and feature-dependent regularizers (see Theorem 2).

Figure 1: An illustration of how two data points, $\mathbf{x}_1$ and $\mathbf{x}_2$, are transformed in mixup (top) and noisy feature mixup (NFM) with $\mathcal{S} := \{0\}$ (bottom).

- We provide empirical results in support of our theoretical findings, showing that NFM improves robustness with respect to various forms of data perturbation across a wide range of state-of-the-art architectures on computer vision benchmark tasks.

In the Supplementary Materials (**SM**), we provide proofs for our theorems along with additional theoretical and empirical results to gain more insights into NFM. In particular, we show that NFM can implicitly increase classification margin (see Proposition 1 in **SM** C) and the noise injection procedure in NFM can robustify manifold mixup in a probabilistic sense (see Theorem 5 in **SM** D). We also provide and discuss generalization bounds for NFM (see Theorem 6 and 7 in **SM** E).

**Notation.** $I$ denotes identity matrix, $[K] := \{1, \ldots, K\}$, the superscript $^T$ denotes transposition, $\circ$ denotes composition, $\odot$ denotes Hadamard product, $\mathbb{1}$ denotes the vector with all components equal one. For a vector $v$, $v^k$ denotes its $k$th component and $\|v\|_p$ denotes its $l_p$ norm for $p > 0$. $conv(\mathcal{X})$ denote the convex hull of $\mathcal{X}$. $M_\lambda(a, b) := \lambda a + (1 - \lambda)b$, for random variables $a, b, \lambda$. $\delta_z$ denotes the Dirac delta function, defined as $\delta_z(x) = 1$ if $x = z$ and $\delta_z(x) = 0$ otherwise. $\mathbb{1}_A$ denotes indicator function of the set $A$. For $\alpha, \beta > 0$, $\tilde{\mathcal{D}}_\lambda := \frac{\alpha}{\alpha+\beta}Beta(\alpha + 1, \beta) + \frac{\beta}{\alpha+\beta}Beta(\beta + 1, \alpha)$ denotes a uniform mixture of two Beta distributions. For two vectors $a, b$, $\cos(a, b) := \langle a, b \rangle / \|a\|_2 \|b\|_2$ denotes their cosine similarity. $\mathcal{N}(a, b)$ is a Gaussian distribution with mean $a$ and covariance $b$.

## 2    RELATED WORK

**Regularization.** Regularization refers to any technique that reduces overfitting in machine learning; see (Mahoney & Orecchia, 2011; Mahoney, 2012) and references therein, in particular for a discussion of *implicit* regularization, a topic that has received attention recently in the context of stochastic gradient optimization applied to neural network models. Traditional regularization techniques such as ridge regression, weight decay and dropout do not make use of the training data to reduce the model capacity. A powerful class of techniques is data augmentation, which constructs additional examples from the training set, e.g., by applying geometric transformations to the original data (Shorten & Khoshgoftaar, 2019). A recently proposed technique is mixup (Zhang et al., 2017), where the examples are created by taking convex combinations of pairs of inputs and their labels. Verma et al. (2019) extends mixup to hidden representations in deep neural networks. Subsequent works by Greenewald et al. (2021); Yin et al. (2021); Engstrom et al. (2019); Kim et al. (2020a); Yun et al. (2019); Hendrycks et al. (2019) introduce different variants and extensions of mixup. Regularization is also intimately connected to robustness (Hoffman et al., 2019; Sokolić et al., 2017; Novak et al., 2018; Elsayed et al., 2018; Moosavi-Dezfooli et al., 2019). Adding to the list is NFM, a powerful regularization method that we propose to improve model robustness.

**Robustness.** Model robustness is an increasingly important issue in modern machine learning. Robustness with respect to adversarial examples (Kurakin et al., 2016) can be achieved by adversarial training (Goodfellow et al., 2014; Madry et al., 2017; Utrera et al., 2020). Several works present theoretical justifications to observed robustness and how data augmentation can improve it (Hein & Andriushchenko, 2017; Yang et al., 2020b; Couellan, 2021; Pinot et al., 2019a; 2021; Zhang et al., 2020; 2021; Carratino et al., 2020; Kimura, 2020; Dao et al., 2019; Wu et al., 2020; Gong et al., 2020; Chen et al., 2020). Relatedly, Fawzi et al. (2016); Franceschi et al. (2018); Lim et al. (2021)

investigate how noise injection can be used to improve robustness. Parallel to this line of work, we provide theory to understand how NFM can improve robustness. Also related is the study of the trade-offs between robustness and accuracy (Min et al., 2020; Zhang et al., 2019; Tsipras et al., 2018; Schmidt et al., 2018; Su et al., 2018; Raghunathan et al., 2020; Yang et al., 2020a).

## 3    NOISY FEATURE MIXUP

Noisy Feature Mixup is a generalization of input mixup (Zhang et al., 2017) and manifold mixup (Verma et al., 2019). *The main novelty of NFM against manifold mixup lies in the injection of noise when taking convex combinations of pairs of input and hidden layer features.* Fig. 1 illustrates, at a high level, how this modification alters the region in which the resulting augmented data resides. Fig. 2 shows that NFM is most effective at smoothing the decision boundary of the trained classifiers; compared to noise injection and mixup alone, it imposes the strongest smoothness on this dataset.

Formally, we consider multi-class classification with $K$ labels. Denote the input space by $\mathcal{X} \subset \mathbb{R}^d$ and the output space by $\mathcal{Y} = \mathbb{R}^K$. The classifier, $g$, is constructed from a learnable map $f : \mathcal{X} \to \mathbb{R}^K$, mapping an input $x$ to its label, $g(x) = \arg\max_k f^k(x) \in [K]$. We are given a training set, $\mathcal{Z}_n := \{(x_i, y_i)\}_{i=1}^n$, consisting of $n$ pairs of input and one-hot label, with each training pair $z_i := (x_i, y_i) \in \mathcal{X} \times \mathcal{Y}$ drawn i.i.d. from a ground-truth distribution $\mathcal{D}$. We consider training a deep neural network $f := f_k \circ g_k$, where $g_k : \mathcal{X} \to g_k(\mathcal{X})$ maps an input to a hidden representation at layer $k$, and $f_k : g_k(\mathcal{X}) \to g_L(\mathcal{X}) := \mathcal{Y}$ maps the hidden representation to a one-hot label at layer $L$. Here, $g_k(\mathcal{X}) \subset \mathbb{R}^{d_k}$ for $k \in [L]$, $d_L := K$, $g_0(x) = x$ and $f_0(x) = f(x)$.

Training $f$ using NFM consists of the following steps:

1. Select a random layer $k$ from a set, $\mathcal{S} \subset \{0\} \cup [L]$, of eligible layers in the neural network.
2. Process two random data minibatches $(x, y)$ and $(x', y')$ as usual, until reaching layer $k$. This gives us two immediate minibatches $(g_k(x), y)$ and $(g_k(x'), y')$.
3. Perform mixup on these intermediate minibatches, producing the mixed minibatch:

$$(\tilde{g}_k, \tilde{y}) := (M_\lambda(g_k(x), g_k(x')), M_\lambda(y, y')), \tag{1}$$

   where the mixing level $\lambda \sim Beta(\alpha, \beta)$, with the hyper-parameters $\alpha, \beta > 0$.
4. Produce noisy mixed minibatch by injecting additive and multiplicative noise:

$$(\tilde{\tilde{g}}_k, \tilde{y}) := ((\mathbb{1} + \sigma_{mult}\xi_k^{mult}) \odot M_\lambda(g_k(x), g_k(x')) + \sigma_{add}\xi_k^{add}, M_\lambda(y, y')), \tag{2}$$

   where the $\xi_k^{add}$ and $\xi_k^{mult}$ are $\mathbb{R}^{d_k}$-valued independent random variables modeling the additive and multiplicative noise respectively, and $\sigma_{add}, \sigma_{mult} \geq 0$ are pre-specified noise levels.
5. Continue the forward pass from layer $k$ until the output using the noisy mixed minibatch $(\tilde{\tilde{g}}_k, \tilde{y})$.
6. Compute the loss and gradients that update all the parameters of the network.

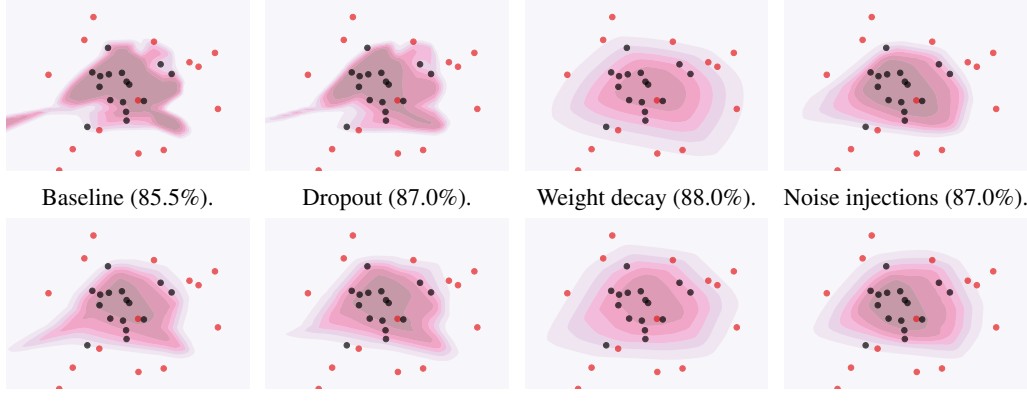

| Baseline (85.5%). | Dropout (87.0%). | Weight decay (88.0%). | Noise injections (87.0%). |

| Mixup (84.5%). | Manifold mixup (88.5%). | Noisy mixup (89.0%). | NFM (90.0%). |

Figure 2: The decision boundaries and test accuracy (in parenthesis) for different training schemes on a toy dataset in binary classification (see Subsection F.2 for details).

At the level of implementation, following (Verma et al., 2019), we backpropagate gradients through the entire computational graph, including those layers before the mixup layer $k$.

In the case where $\sigma_{add} = \sigma_{mult} = 0$, NFM reduces to manifold mixup (Verma et al., 2019). If in addition $\mathcal{S} = \{0\}$, it reduces to the original mixup method (Zhang et al., 2017). The main difference between NFM and manifold mixup lies in the noise injection of the fourth step above. Note that NFM is equivalent to injecting noise into $g_k(x), g_k(x')$ first, then performing mixup on the resulting pair, i.e., the order that the third and fourth steps occur does not change the resulting noisy mixed minibatch. For simplicity, we have used the same mixing level, noise distribution, and noise levels for all layers in $\mathcal{S}$ in our formulation.

Within the above setting, we consider the expected NFM loss:

$$L^{NFM}(f) = \mathbb{E}_{(x,y),(x',y')\sim\mathcal{D}}\mathbb{E}_{k\sim\mathcal{S}}\mathbb{E}_{\lambda\sim Beta(\alpha,\beta)}\mathbb{E}_{\boldsymbol{\xi}_k\sim\mathcal{Q}}l(f_k(M_{\lambda,\boldsymbol{\xi}_k}(g_k(x), g_k(x'))), M_\lambda(y, y')),$$

where $l : \mathbb{R}^K \times \mathbb{R}^K \to [0, \infty)$ is a loss function (note that here we have suppressed the dependence of both $l$ and $f$ on the learnable parameter $\theta$ in the notation), $\boldsymbol{\xi}_k := (\xi_k^{add}, \xi_k^{mult})$ are drawn from some probability distribution $\mathcal{Q}$ with finite first two moments, and

$$M_{\lambda,\boldsymbol{\xi}_k}(g_k(x), g_k(x')) := (\mathbb{1} + \sigma_{mult}\xi_k^{mult}) \odot M_\lambda(g_k(x), g_k(x')) + \sigma_{add}\xi_k^{add}.$$

NFM seeks to minimize a stochastic approximation of $L^{NFM}(f)$ by sampling a finite number of $k, \lambda, \boldsymbol{\xi}_k$ values and using minibatch gradient descent to minimize this loss approximation.

## 4 THEORY

In this section, we provide mathematical analysis to understand NFM. We begin with formulating NFM in the framework of vicinal risk minimization and interpreting NFM as a stochastic learning strategy in Subsection 4.1. Next, we study NFM via the lens of implicit regularization in Subsection 4.2. Our key contribution is Theorem 1, which shows that minimizing the NFM loss function is approximately equivalent to minimizing a sum of the original loss and feature-dependent regularizers, amplifying the regularizing effects of manifold mixup and noise injection according to the mixing and noise levels. In Subsection 4.3, we focus on demonstrating how NFM can enhance model robustness via the lens of distributionally robust optimization. The key result of Theorem 2 shows that NFM loss is approximately the upper bound on a regularized version of an adversarial loss, and thus training with NFM not only improves robustness but can also mitigate robust over-fitting, a dominant phenomenon where the robust test accuracy starts to decrease during training (Rice et al., 2020).

### 4.1 NFM: BEYOND EMPIRICAL RISK MINIMIZATION

The standard approach in statistical learning theory (Bousquet et al., 2003) is to select a hypothesis function $f : \mathcal{X} \to \mathcal{Y}$ from a pre-defined hypothesis class $\mathcal{F}$ to minimize the expected risk with respect to $\mathcal{D}$ and to solve the risk minimization problem: $\inf_{f\in\mathcal{F}} \mathcal{R}(f) := \mathbb{E}_{(x,y)\sim\mathcal{D}}[l(f(x), y)]$, for a suitable choice of loss function $l$. In practice, we do not have access to the ground-truth distribution. Instead, we find an approximate solution by solving the empirical risk minimization (ERM) problem, in which case $\mathcal{D}$ is approximated by the empirical distribution $\mathbb{P}_n = \frac{1}{n}\sum_{i=1}^n \delta_{z_i}$. In other words, in ERM we solve the problem: $\inf_{f\in\mathcal{F}} \mathcal{R}_n(f) := \frac{1}{n}\sum_{i=1}^n l(f(x_i), y_i)$.

However, when the training set is small or the model capacity is large (as is the case for deep neural networks), ERM may suffer from overfitting. Vicinal risk minimization (VRM) is a data augmentation principle introduced in (Vapnik, 2013) that goes beyond ERM, aiming to better estimate expected risk and reduce overfitting. In VRM, a model is trained not simply on the training set, but on samples drawn from a vicinal distribution, that smears the training data to their vicinity. With appropriate choices for this distribution, the VRM approach has resulted in several effective regularization schemes (Chapelle et al., 2001). Input mixup (Zhang et al., 2017) can be viewed as an example of VRM, and it turns out that NFM can be constructed within a VRM framework at the feature level (see Section A in **SM**). On a high level, NFM can be interpreted as a random procedure that introduces feature-dependent noise into the layers of the deep neural network. Since the noise injections are applied only during training and not inference, NFM is an instance of a stochastic learning strategy. Note that the injection strategy of NFM differs from those of An (1996); Camuto et al. (2020); Lim

et al. (2021). Here, the structure of the injected noise differs from iteration to iteration (based on the layer chosen) and depends on the training data in a different way. We expect NFM to amplify the benefits of training using either noise injection or mixup alone, as will be shown next.

## 4.2 Implicit Regularization of NFM

We consider loss functions of the form $l(f(x), y) := h(f(x)) - yf(x)$, which includes standard choices such as the logistic loss and the cross-entropy loss, and recall that $f := f_k \circ g_k$. Denote $L_n^{std} := \frac{1}{n} \sum_{i=1}^n l(f(x_i), y_i)$ and let $\mathcal{D}_x$ be the empirical distribution of training samples $\{x_i\}_{i \in [n]}$. We shall show that NFM exhibits a natural form of implicit regularization, i.e., regularization imposed implicitly by the stochastic learning strategy, without explicitly modifying the loss.

Let $\epsilon > 0$ be a small parameter. In the sequel, we rescale $1 - \lambda \mapsto \epsilon(1 - \lambda)$, $\sigma_{add} \mapsto \epsilon\sigma_{add}$, $\sigma_{mult} \mapsto \epsilon\sigma_{mult}$, and denote $\nabla_k f$ and $\nabla_k^2 f$ as the first and second directional derivative of $f_k$ with respect to $g_k$ respectively, for $k \in \mathcal{S}$. By working in the small parameter regime, we can relate the NFM empirical loss $L_n^{NFM}$ to the original loss $L_n^{std}$ and identify the regularizing effects of NFM.

**Theorem 1.** *Let $\epsilon > 0$ be a small parameter, and assume that $h$ and $f$ are twice differentiable. Then, $L_n^{NFM} = \mathbb{E}_{k \sim \mathcal{S}} L_n^{NFM(k)}$, where*

$$L_n^{NFM(k)} = L_n^{std} + \epsilon R_1^{(k)} + \epsilon^2 \tilde{R}_2^{(k)} + \epsilon^2 \tilde{R}_3^{(k)} + \epsilon^2 \varphi(\epsilon), \tag{3}$$

*with $\tilde{R}_2^{(k)} = R_2^{(k)} + \sigma_{add}^2 R_2^{add(k)} + \sigma_{mult}^2 R_2^{mult(k)}$ and $\tilde{R}_3^{(k)} = R_3^{(k)} + \sigma_{add}^2 R_3^{add(k)} + \sigma_{mult}^2 R_3^{mult(k)}$, where*

$$R_2^{add(k)} = \frac{1}{2n} \sum_{i=1}^n h''(f(x_i)) \nabla_k f(g_k(x_i))^T \mathbb{E}_{\boldsymbol{\xi}_k}[\xi_k^{add}(\xi_k^{add})^T] \nabla_k f(g_k(x_i)), \tag{4}$$

$$R_2^{mult(k)} = \frac{1}{2n} \sum_{i=1}^n h''(f(x_i)) \nabla_k f(g_k(x_i))^T (\mathbb{E}_{\boldsymbol{\xi}_k}[\xi_k^{mult}(\xi_k^{mult})^T] \odot g_k(x_i) g_k(x_i)^T) \nabla_k f(g_k(x_i)), \tag{5}$$

$$R_3^{add(k)} = \frac{1}{2n} \sum_{i=1}^n (h'(f(x_i)) - y_i) \mathbb{E}_{\boldsymbol{\xi}_k}[(\xi_k^{add})^T \nabla_k^2 f(g_k(x_i)) \xi_k^{add}], \tag{6}$$

$$R_3^{mult(k)} = \frac{1}{2n} \sum_{i=1}^n (h'(f(x_i)) - y_i) \mathbb{E}_{\boldsymbol{\xi}_k}[(\xi_k^{mult} \odot g_k(x_i))^T \nabla_k^2 f(g_k(x_i)) (\xi_k^{mult} \odot g_k(x_i))]. \tag{7}$$

*Here, $R_1^k$, $R_2^k$ and $R_3^k$ are the regularizers associated with the loss of manifold mixup (see Theorem 3 in **SM** for their explicit expression), and $\varphi$ is some function such that $\lim_{\epsilon \to 0} \varphi(\epsilon) = 0$.*

Theorem 1 implies that, when compared to manifold mixup, NFM introduces additional smoothness, regularizing the directional derivatives, $\nabla_k f(g_k(x_i))$ and $\nabla_k^2 f(g_k(x_i))$, with respect to $g_k(x_i)$, according to the noise levels $\sigma_{add}$ and $\sigma_{mult}$, and amplifying the regularizing effects of manifold mixup and noise injection. In particular, making $\nabla^2 f(x_i)$ small can lead to smooth decision boundaries (at the input level), while reducing the confidence of model predictions. On the other hand, making the $\nabla_k f(g_k(x_i))$ small can lead to improvement in model robustness, which we discuss next.

## 4.3 Robustness of NFM

We show that NFM improves model robustness. We do this by considering the following three lenses: (1) implicit regularization and classification margin; (2) distributionally robust optimization; and (3) a probabilistic notion of robustness. We focus on (2) in the main paper. See Section C-D in **SM** and the last paragraph in this subsection for details on (1) and (3).

We now demonstrate how NFM helps adversarial robustness. By extending the analysis of Zhang et al. (2017); Lamb et al. (2019), we can relate the NFM loss function to the one used for adversarial training, which can be viewed as an instance of distributionally robust optimization (DRO) (Kwon et al., 2020; Kuhn et al., 2019; Rahimian & Mehrotra, 2019) (see also Proposition 3.1 in (Staib & Jegelka, 2017)). DRO provides a framework for local worst-case risk minimization, minimizing supremum of the risk in an ambiguity set, such as in the vicinity of the empirical data distribution.

Following (Lamb et al., 2019), we consider the binary cross-entropy loss, setting $h(z) = \log(1 + e^z)$, with the labels $y$ taking value in $\{0, 1\}$ and the classifier model $f : \mathbb{R}^d \to \mathbb{R}$. In the following, we assume that the model parameter $\theta \in \Theta := \{\theta : y_i f(x_i) + (y_i - 1)f(x_i) \geq 0 \text{ for all } i \in [n]\}$. Note that this set contains the set of all parameters with correct classifications of training samples (before applying NFM), since $\{\theta : 1_{\{f(x_i) \geq 0\}} = y_i \text{ for all } i \in [n]\} \subset \Theta$. Therefore, the condition of $\theta \in \Theta$ is satisfied when the model classifies all labels correctly for the training data before applying NFM. Since, in practice, the training error often becomes zero in finite time, we study the effect of NFM on model robustness in the regime of $\theta \in \Theta$.

Working in the data-dependent parameter space $\Theta$, we have the following result.

**Theorem 2.** *Let* $\theta \in \Theta := \{\theta : y_i f(x_i) + (y_i - 1)f(x_i) \geq 0 \text{ for all } i \in [n]\}$ *such that* $\nabla_k f(g_k(x_i))$ *and* $\nabla_k^2 f(g_k(x_i))$ *exist for all* $i \in [n]$, $k \in \mathcal{S}$. *Assume that* $f_k(g_k(x_i)) = \nabla_k f(g_k(x_i))^T g_k(x_i)$, $\nabla_k^2 f(g_k(x_i)) = 0$ *for all* $i \in [n]$, $k \in \mathcal{S}$. *In addition, suppose that* $\|\nabla f(x_i)\|_2 > 0$ *for all* $i \in [n]$, $\mathbb{E}_{r \sim \mathcal{D}_x}[g_k(r)] = 0$ *and* $\|g_k(x_i)\|_2 \geq c_x^{(k)} \sqrt{d_k}$ *for all* $i \in [n]$, $k \in \mathcal{S}$. *Then,*

$$L_n^{NFM} \geq \frac{1}{n} \sum_{i=1}^n \max_{\|\delta_i\|_2 \leq \epsilon_i^{mix}} l(f(x_i + \delta_i), y_i) + L_n^{reg} + \epsilon^2 \phi(\epsilon), \tag{8}$$

*where* $\epsilon_i^{mix} := \epsilon \mathbb{E}_{\lambda \sim \tilde{\mathcal{D}}_\lambda}[1 - \lambda] \cdot \mathbb{E}_{k \sim \mathcal{S}}\left[r_i^{(k)} c_x^{(k)} \frac{\|\nabla_k f(g_k(x_i))\|_2}{\|\nabla f(x_i)\|_2} \sqrt{d_k}\right]$ *and* $L_n^{reg} := \frac{1}{2n} \sum_{i=1}^n |h''(f(x_i))|(\epsilon_i^{reg})^2$, *with* $r_i^{(k)} := |\cos(\nabla_k f(g_k(x_i)), g_k(x_i))|$ *and*

$$(\epsilon_i^{reg})^2 := \epsilon^2 \|\nabla_k f(g_k(x_i))\|_2^2 \Big( \mathbb{E}_\lambda[(1 - \lambda)]^2 \mathbb{E}_{x_r}[\|g_k(x_r)\|_2^2 \cos(\nabla_k f(g_k(x_i)), g_k(x_r))^2]$$

$$+ \sigma_{add}^2 \mathbb{E}_{\xi_k}[\|\xi_k^{add}\|_2^2 \cos(\nabla_k f(g_k(x_i)), \xi_k^{add})^2]$$

$$+ \sigma_{mult}^2 \mathbb{E}_{\xi_k}[\|\xi_k^{mult} \odot g_k(x_i)\|_2^2 \cos(\nabla_k f(g_k(x_i)), \xi_k^{mult} \odot g_k(x_i))^2] \Big), \tag{9}$$

*and* $\phi$ *is some function such that* $\lim_{\epsilon \to 0} \phi(\epsilon) = 0$.

The second assumption stated in Theorem 2 is similar to the one made in Lamb et al. (2019); Zhang et al. (2020), and is satisfied by linear models and deep neural networks with ReLU activation function and max-pooling. Theorem 2 shows that the NFM loss is approximately an upper bound of the adversarial loss with $l_2$ attack of size $\epsilon^{mix} = \min_{i \in [n]} \epsilon_i^{mix}$, plus a feature-dependent regularization term $L_n^{reg}$ (see **SM** for further discussions). Therefore, we see that minimizing the NFM loss not only results in a small adversarial loss, while retaining the robustness benefits of manifold mixup, but it also imposes additional smoothness, due to noise injection, on the adversarial loss. The latter can help mitigate robust overfitting and improve test performance (Rice et al., 2020; Rebuffi et al., 2021).

NFM can also implicitly increase the classification margin (see Section C of **SM**). Moreover, since the main novelty of NFM lies in the introduction of noise injection, it would be insightful to isolate the robustness boosting benefits of injecting noise on top of manifold mixup. We demonstrate these advantages via the lens of probabilistic robustness in Section D of **SM**.

## 5 EMPIRICAL RESULTS

In this section, we study the test performance of models trained with NFM, and examine to what extent NFM can improve robustness to input perturbations. We demonstrate the tradeoff between predictive accuracy on clean and perturbed test sets. We consider input perturbations that are common in the literature: (a) white noise; (b) salt and pepper; and (c) adversarial perturbations (see Section F).

We evaluate the average performance of NFM with different model architectures on CIFAR-10 (Krizhevsky, 2009), CIFAR-100 (Krizhevsky, 2009), ImageNet (Deng et al., 2009), and CIFAR-10c (Hendrycks & Dietterich, 2019). We use a pre-activated residual network (ResNet) with depth 18 (He et al., 2016) on small scale tasks. For more challenging tasks, we consider the performance of wide ResNet-18 (Zagoruyko & Komodakis, 2016) and ResNet-50 architectures, respectively.

**Baselines.** We evaluate against related data augmentation schemes that have shown performance improvements in recent years: mixup (Zhang et al., 2017); manifold mixup (Verma et al., 2019);

cutmix (Yun et al., 2019); puzzle mixup (Kim et al., 2020b); and noisy mixup (Yang et al., 2020b). Further, we compare to vanilla models trained without data augmentation (baseline), models trained with label smoothing, and those trained on white noise perturbed inputs.

**Experimental details.** All hyperparameters are consistent with those of the baseline model across the ablation experiments. In the models trained on the different data augmentation schemes, we keep $\alpha$ fixed, i.e., the parameter defining $Beta(\alpha, \alpha)$, from which the $\lambda$ parameter controlling the convex combination between data point pairs is sampled. Across all models trained with NFM, we control the level of noise injections by fixing the additive noise level to $\sigma_{add} = 0.4$ and multiplicative noise to $\sigma_{mult} = 0.2$. To demonstrate the significant improvements on robustness upon the introduction of these small input perturbations, we show a second model ('*') that was injected with higher noise levels (i.e., $\sigma_{add} = 1.0$, $\sigma_{mult} = 0.5$). See **SM** (Section F.5) for further details and comparisons against NFM models trained on various other levels of noise injections.

### 5.1 CIFAR10

**Pre-activated ResNet-18.** Table 1 summarizes the performance improvements and indicates a consistent robustness across different $\alpha$ values. The model trained with NFM outperforms the baseline model on the clean test set, while being more robust to input perturbations (Fig. 3; left). This advantage is also displayed in the models trained with mixup and manifold mixup, though in a less pronounced way. Notably, the NFM model is also robust to salt and pepper perturbations and could be significantly more so by further increasing the noise levels (Fig. 3; right).

### 5.2 CIFAR-100

**Wide ResNet-18.** Previous work indicates that data augmentation has a positive effect on performance for this dataset (Zhang et al., 2017). Fig. 4 (left) confirms that mixup and manifold mixup improve the generalization performance on clean data and highlights the advantage of data augmentation. The NFM training scheme is also capable of further improving the generalization performance. In

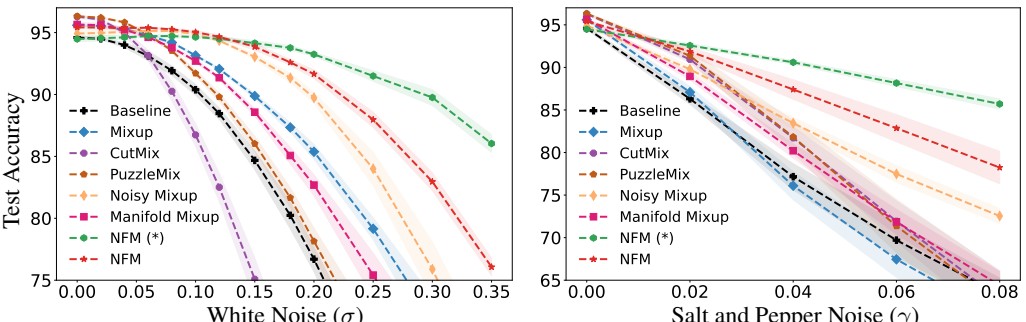

Figure 3: Pre-actived ResNet-18 evaluated on CIFAR-10 with different training schemes. Shaded regions indicate one standard deviation about the mean. Averaged across 5 random seeds.

Table 1: Robustness of ResNet-18 w.r.t. white noise ($\sigma$) and salt and pepper ($\gamma$) perturbations evaluated on CIFAR-10. The results are averaged over 5 models trained with different seed values.

| Scheme | Clean (%) | $\sigma$ (%) | | | $\gamma$ (%) | | |
|---|---|---|---|---|---|---|---|
| | | 0.1 | 0.2 | 0.3 | 0.02 | 0.04 | 0.1 |
| Baseline | 94.6 | 90.4 | 76.7 | 56.3 | 86.3 | 76.1 | 55.2 |
| Baseline + Noise | 94.4 | 94.0 | 87.5 | 71.2 | 89.3 | 82.5 | 64.9 |
| Baseline + Label Smoothing | 95.0 | 91.3 | 77.5 | 56.9 | 87.7 | 79.2 | 60.0 |
| Mixup ($\alpha = 1.0$) Zhang et al. (2017) | 95.6 | 93.2 | 85.4 | 71.8 | 87.1 | 76.1 | 55.2 |
| CutMix Yun et al. (2019) | **96.3** | 86.7 | 60.8 | 32.4 | 90.9 | 81.7 | 54.7 |
| PuzzleMix Kim et al. (2020b) | **96.3** | 91.7 | 78.1 | 59.9 | 91.4 | 81.8 | 54.4 |
| Manifold Mixup ($\alpha = 1.0$) Verma et al. (2019) | 95.7 | 92.7 | 82.7 | 67.6 | 88.9 | 80.2 | 57.6 |
| Noisy Mixup ($\alpha = 1.0$) Yang et al. (2020b) | 78.9 | 78.6 | 66.6 | 46.7 | 66.6 | 53.4 | 25.9 |
| Noisy Feature Mixup ($\alpha = 1.0$) | 95.4 | **95.0** | **91.6** | **83.0** | **91.9** | **87.4** | **73.3** |

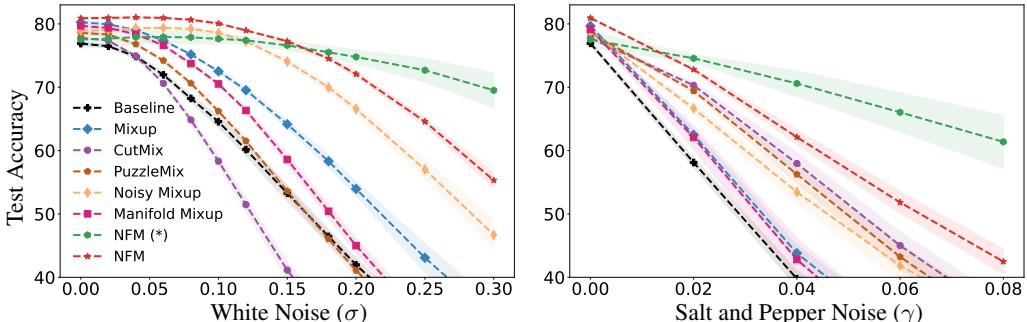

Figure 4: Wide ResNets evaluated on CIFAR-100. Averaged across 5 random seeds.

Table 2: Robustness of Wide-ResNet-18 w.r.t. white noise ($\sigma$) and salt and pepper ($\gamma$) perturbations evaluated on CIFAR-100. The results are averaged over 5 models trained with different seed values.

| Scheme | Clean (%) | $\sigma$ (%) | | | $\gamma$ (%) | | |
|---|---|---|---|---|---|---|---|
| | | 0.1 | 0.2 | 0.3 | 0.02 | 0.04 | 0.1 |
| Baseline | 76.9 | 64.6 | 42.0 | 23.5 | 58.1 | 39.8 | 15.1 |
| Baseline + Noise | 76.1 | 75.2 | 60.5 | 37.6 | 64.9 | 51.3 | 23.0 |
| Mixup ($\alpha = 1.0$) Zhang et al. (2017) | 80.3 | 72.5 | 54.0 | 33.4 | 62.5 | 43.8 | 16.2 |
| CutMix Yun et al. (2019) | 77.8 | 58.3 | 28.1 | 13.8 | 70.3 | 58. | 24.8 |
| PuzzleMix (200 epochs) Kim et al. (2020b) | 78.6 | 66.2 | 41.1 | 22.6 | 69.4 | 56.3 | 23.3 |
| PuzzleMix (1200 epochs) Kim et al. (2020b) | 80.3 | 53.0 | 19.1 | 6.2 | 69.3 | 51.9 | 15.7 |
| Manifold Mixup ($\alpha = 1.0$) Verma et al. (2019) | 79.7 | 70.5 | 45.0 | 23.8 | 62.1 | 42.8 | 14.8 |
| Noisy Mixup ($\alpha = 1.0$) Yang et al. (2020b) | 78.9 | 78.6 | 66.6 | 46.7 | 66.6 | 53.4 | 25.9 |
| Noisy Feature Mixup ($\alpha = 1.0$) | **80.9** | **80.1** | **72.1** | **55.3** | **72.8** | **62.1** | **34.4** |

Table 3: Robustness of ResNet-50 w.r.t. white noise ($\sigma$) and salt and pepper ($\gamma$) perturbations evaluated on ImageNet. Here, the NFM training scheme improves both the predictive accuracy on clean data and robustness with respect to data perturbations.

| Scheme | Clean (%) | $\sigma$ (%) | | | $\gamma$ (%) | | |
|---|---|---|---|---|---|---|---|
| | | 0.1 | 0.25 | 0.5 | 0.06 | 0.1 | 0.15 |
| Baseline | 76.0 | 73.5 | 67.0 | 50.1 | 53.2 | 50.4 | 45.0 |
| Manifold Mixup ($\alpha = 0.2$) Verma et al. (2019) | 76.7 | 74.9 | 70.3 | 57.5 | 58.1 | 54.6 | 49.5 |
| Noisy Feature Mixup ($\alpha = 0.2$) | **77.0** | **76.5** | **72.0** | **60.1** | 58.3 | 56.0 | 52.3 |
| Noisy Feature Mixup ($\alpha = 1.0$) | 76.8 | 76.2 | 71.7 | 60.0 | **60.9** | **58.8** | **54.4** |

addition, we see that the model trained with NFM is less sensitive to both white noise and salt and pepper perturbations. These results are surprising, as robustness is often thought to be at odds with accuracy (Tsipras et al., 2018). However, we demonstrate NFM has the ability to improve both accuracy and robustness. Table 2 indicates that for the same $\alpha$, NFM can achieve an average test accuracy of 80.9% compared to only 80.3% in the mixup setting.

## 5.3 IMAGENET

**ResNet-50.** Table 3 similarly shows that NFM improves both the generalization and robustness capacities with respect to data perturbations. Although less pronounced in comparison to previous datasets, NFM shows a favorable trade-off without requiring additional computational resources. Note that due to computational costs, we do not average across multiple seeds and only compare NFM to the baseline and manifold mixup models.

## 5.4 CIFAR-10C

In Figure 6 we use the CIFAR-10C dataset (Hendrycks & Dietterich, 2019) to demonstrate that models trained with NFM are more robust to a range of perturbations on natural images. Figure 6 (left) shows

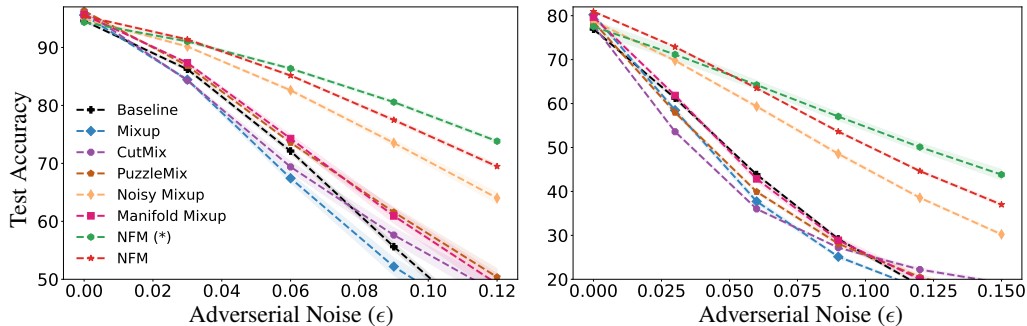

Figure 5: Pre-actived ResNet-18 evaluated on CIFAR-10 (left) and Wide ResNet-18 evaluated on CIFAR-100 (right) with respect to adversarially perturbed inputs.

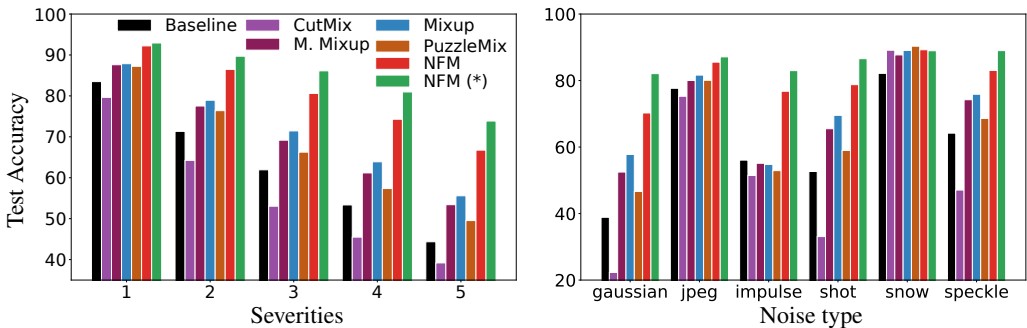

Figure 6: Pre-actived ResNet-18 evaluated on CIFAR-10c.

the average test accuracy across six selected perturbations and demonstrates the advantage of NFM being particularly pronounced with the progression of severity levels. The right figure shows the performance on the same set of six perturbations for the median severity level 3. NFM excels on Gaussian, impulse, speckle and shot noise, and is competitive with the rest on the snow perturbation.

## 5.5 ROBUSTNESS TO ADVERSARIAL EXAMPLES

So far we have only considered white noise and salt and pepper perturbations. We further consider adversarial perturbations. Here, we use projected gradient decent (Madry et al., 2017) with 7 iterations and various $\epsilon$ levels to construct the adversarial perturbations. Fig. 5 highlights the improved resilience of ResNets trained with NFM to adversarial input perturbations and shows this consistently on both CIFAR-10 (left) and CIFAR-100 (right). Models trained with both mixup and manifold mixup do not show a substantially increased resilience to adversarial perturbations.

In Section F.6, we compare NFM to models that are adversarially trained. There, we see that adversarially trained models are indeed more robust to adversarial attacks, while at the same time being less accurate on clean data. However, models trained with NFM show an advantage compared to adversarially trained models when faced with salt and pepper perturbations.

## 6 CONCLUSION

We introduce Noisy Feature Mixup, an effective data augmentation method that combines mixup and noise injection. We identify the implicit regularization effects of NFM, showing that the effects are amplifications of those of manifold mixup and noise injection. Moreover, we demonstrate the benefits of NFM in terms of superior model robustness, both theoretically and experimentally. Our work inspires a range of interesting future directions, including theoretical investigations of the trade-offs between accuracy and robustness for NFM and applications of NFM beyond computer vision tasks. Further, it will be interesting to study whether NFM may also lead to better model calibration by extending the analysis of Thulasidasan et al. (2019); Zhang et al. (2021).

## CODE OF ETHICS

We acknowledge that we have read and commit to adhering to the ICLR Code of Ethics.

## REPRODUCIBILITY

The codes that can be used to reproduce the empirical results, as well as description of the data processing steps, presented in this paper are available as a zip file in Supplementary Material at OpenReview.net. The codes are also available at `https://github.com/erichson/NFM`. For the theoretical results, all assumptions, proofs and the related discussions are provided in **SM**.

## ACKNOWLEDGMENTS

S. H. Lim would like to acknowledge the WINQ Fellowship and the Knut and Alice Wallenberg Foundation for providing support of this work. N. B. Erichson and M. W. Mahoney would like to acknowledge IARPA (contract W911NF20C0035), NSF, and ONR for providing partial support of this work. Our conclusions do not necessarily reflect the position or the policy of our sponsors, and no official endorsement should be inferred. We are also grateful for the generous support from Amazon AWS.

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

# Supplementary Material (SM) for "Noisy Feature Mixup"

**Organizational Details.** This **SM** is organized as follows.

- In Section A, we study the regularizing effects of NFM within the vicinal risk minimization framework, relating the effects to those of mixup and noise injection.
- In Section B, we restate the results presented in the main paper and provide their proof.
- In Section C, we study robutsness of NFM through the lens of implicit regularization, showing that NFM can implicitly increase the classification margin.
- In Section D, we study robustness of NFM via the lens of probabilistic robustness, showing that noise injection can improve robustness on top of manifold mixup while keeping track of maximal loss in accuracy incurred under attack by tuning the noise levels.
- In Section E, we provide results on generalization bounds for NFM and their proofs, identifying the mechanisms by which NFM can lead to improved generalization bound.
- In Section F, we provide additional experimental results and their details.

We recall the notation that we use in the main paper as well as this **SM**.

**Notation.** $I$ denotes identity matrix, $[K] := \{1, \ldots, K\}$, the superscript $^T$ denotes transposition, $\circ$ denotes composition, $\odot$ denotes Hadamard product, $\mathbb{1}$ denotes the vector with all components equal one. For a vector $v$, $v^k$ denotes its $k$th component and $\|v\|_p$ denotes its $l_p$ norm for $p > 0$. $conv(\mathcal{X})$ denote the convex hull of $\mathcal{X}$. $M_\lambda(a, b) := \lambda a + (1 - \lambda)b$, for random variables $a, b, \lambda$. $\delta_z$ denotes the Dirac delta function, defined as $\delta_z(x) = 1$ if $x = z$ and $\delta_z(x) = 0$ otherwise. $\mathbb{1}_A$ denotes indicator function of the set $A$. For $\alpha, \beta > 0$, $\tilde{\mathcal{D}}_\lambda := \frac{\alpha}{\alpha+\beta} Beta(\alpha + 1, \beta) + \frac{\beta}{\alpha+\beta} Beta(\beta + 1, \alpha)$, a uniform mixture of two Beta distributions. For two vectors $a, b$, $\cos(a, b) := \langle a, b \rangle / \|a\|_2 \|b\|_2$ denotes their cosine similarity. $\mathcal{N}(a, b)$ denotes the Gaussian distribution with mean $a$ and covariance $b$.

## A  NFM Through the Lens of Vicinal Risk Minimization

In this section, we shall show that NFM can be constructed within a vicinal risk minimization (VRM) framework at the level of both input and hidden layer representations.

To begin with, we define a class of vicinal distributions and then relate NFM to such distributions.

**Definition 1** (Randomly perturbed feature distribution). *Let $\mathcal{Z}_n = \{z_1, \ldots, z_n\}$ be a feature set. We say that $\mathbb{P}'_n$ is an $e_i$-randomly perturbed feature distribution if there exists a set $\{z'_1, \ldots, z'_n\}$ such that $\mathbb{P}'_n = \frac{1}{n} \sum_{i=1}^n \delta_{z'_i}$, with $z'_i = z_i + e_i$, for some random variable $e_i$ (possibly dependent on $\mathcal{Z}_n$) drawn from a probability distribution.*

Note that the support of an $e_i$-randomly perturbed feature distribution may be larger than that of $\mathcal{Z}$.

If $\mathcal{Z}_n$ is an input dataset and the $e_i$ are bounded variables such that $\|e_i\| \leq \beta$ for some $\beta \geq 0$, then $\mathbb{P}'_n$ is a $\beta$-locally perturbed data distribution according to Definition 2 in (Kwon et al., 2020). Examples of $\beta$-locally perturbed data distribution include that associated with denoising autoencoder, input mixup, and adversarial training (see Example 1-3 in (Kwon et al., 2020)). Definition 1 can be viewed as an extension of the definition in (Kwon et al., 2020), relaxing the boundedness condition on the $e_i$ to cover a wide families of perturbed feature distribution. One simple example is the Gaussian distribution, i.e., when $e_i \sim \mathcal{N}(0, \sigma_i^2)$, which models Gaussian noise injection into the features. Another example is the distribution associated with NFM, which we now discuss.

To keep the randomly perturbed distribution close to the original distribution, the amplitude of the perturbation should be small. In the sequel, we let $\epsilon > 0$ be a small parameter and rescale $1 - \lambda \mapsto \epsilon(1 - \lambda)$, $\sigma_{add} \mapsto \epsilon\sigma_{add}$ and $\sigma_{mult} \mapsto \epsilon\sigma_{mult}$.

Let $\mathcal{F}_k$ be the family of mappings from $g_k(\mathcal{X})$ to $\mathcal{Y}$ and consider the VRM:

$$\inf_{f_k \in \mathcal{F}_k} \mathcal{R}_n(f_k) := \mathbb{E}_{(g'_k(x), y') \sim \mathbb{P}_n^{(k)}}[l(f_k(g'_k(x))), y')], \tag{10}$$

where $\mathbb{P}_n^{(k)} = \frac{1}{n} \sum_{i=1}^n \delta_{(g'_k(x_i), y'_i)}$, with $g'_k(x_i) = g_k(x_i) + \epsilon e_i^{NFM(k)}$ and $y'_i = y_i + \epsilon e_i^y$, for some random variables $e_i^{NFM(k)}$ and $e_i^y$.

In NFM, we approximate the ground-truth distribution $\mathcal{D}$ using the family of distributions $\{\mathbb{P}_n^{(k)}\}_{k \in \mathcal{S}}$, with a particular choice of $(e_i^{NFM(k)}, e_i^y)$. In the sequel, we denote NFM at the level of $k$th layer as $NFM(k)$ (i.e., the particular case when $\mathcal{S} := \{k\}$).

The following lemma identifies the $(e_i^{NFM(k)}, e_i^y)$ associated with $NFM(k)$ and relates the effects of $NFM(k)$ to those of mixup and noise injection, for any perturbation level $\epsilon > 0$.

**Lemma 1.** *Let $\epsilon > 0$ and denote $z_i(k) := g_k(x_i)$. Learning the neural network map $f$ using $NFM(k)$ is a VRM with the $(\epsilon e_i^{NFM(k)}, \epsilon e_i^y)$-randomly perturbed feature distribution, $\mathbb{P}_n^{(k)} = \frac{1}{n} \sum_{i=1}^n \delta_{(z_i'(k), y_i')}$, with $z_i'(k) := z_i(k) + \epsilon e_i^{NFM(k)}$, $y_i' := y_i + \epsilon e_i^y$, as the vicinal distribution. Here, $e_i^y = (1-\lambda)(\tilde{y}_i - y_i)$,*

$$e_i^{NFM(k)} = (\mathbb{1} + \epsilon \sigma_{mult} \xi_{mult}) \odot e_i^{mixup(k)} + e_i^{noise(k)}, \tag{11}$$

*where $e_i^{mixup(k)} = (1-\lambda)(\tilde{z}_i(k) - z_i(k))$, and $e_i^{noise(k)} = \sigma_{mult} \xi_{mult} \odot z_i(k) + \sigma_{add} \xi_{add}$, with $z_i(k), \tilde{z}_i(k) \in g_k(\mathcal{X})$, $\lambda \sim Beta(\alpha, \beta)$ and $y_i, \tilde{y}_i \in \mathcal{Y}$. Here, $(\tilde{z}_i(k), \tilde{y}_i)$ are drawn randomly from the training set.*

Therefore, the random perturbation associated to NFM is data-dependent, and it consists of a randomly weighted sum of that from injecting noise into the feature and that from mixing pairs of feature samples. As a simple example, one can take $\xi_{add}, \xi_{mult}$ to be independent standard Gaussian random variables, in which case we have $e_i^{noise(k)} \sim \mathcal{N}(0, \sigma_{add}^2 I + \sigma_{mult}^2 diag(z_i(k))^2)$, and $e_i \sim \mathcal{N}(0, \sigma_{add}^2 + \sigma_{mult}^2 M_\lambda(z_i(k), \tilde{z}_i(k))^2)$ in Lemma 1.

We now prove Lemma 1.

*Proof of Lemma 1.* Let $k$ be given and set $\epsilon = 1$ without loss of generality. For every $i \in [n]$, $NFM(k)$ injects noise on top of a mixed sample $z_i'(k)$ and outputs:

$$z_i''(k) = (\mathbb{1} + \sigma_{mult} \xi_{mult}) \odot z_i'(k) + \sigma_{add} \xi_{add} \tag{12}$$

$$= (\mathbb{1} + \sigma_{mult} \xi_{mult}) \odot (\lambda z_i(k) + (1-\lambda)\tilde{z}_i(k)) + \sigma_{add} \xi_{add} \tag{13}$$

$$= z_i(k) + e_i^{NFM(k)}, \tag{14}$$

where $e_i^{NFM(k)} = (1-\lambda)(\tilde{z}_i(k) - z_i(k)) + \sigma_{mult} \xi_{mult} \odot (\lambda z_i(k) + (1-\lambda)\tilde{z}_i(k)) + \sigma_{add} \xi_{add}$.

Now, note that applying mixup to the pair $(z_i(k), \tilde{z}_i(k))$ results in $z_i'(k) = z_i(k) + e_i^{mixup(k)}$, with $e_i^{mixup(k)} = (1-\lambda)(\tilde{z}_i(k) - z_i(k))$, where $z_i(k), \tilde{z}_i(k) \in g_k(\mathcal{X})$ and $\lambda \sim Beta(\alpha, \beta)$, whereas applying noise injection to $z_i(k)$ results in $(\mathbb{1} + \sigma_{mult} \xi_{mult}) \odot z_i(k) + \sigma_{add} \xi_{add} = z_i(k) + e_i^{noise(k)}$, with $e_i^{noise(k)} = \sigma_{mult} \xi_{mult} \odot z_i(k) + \sigma_{add} \xi_{add}$. Rewriting $e_i^{NFM(k)}$ in terms of $e_i^{mixup(k)}$ and $e_i^{noise(k)}$ gives

$$e_i^{NFM(k)} = (\mathbb{1} + \sigma_{mult} \xi_{mult}) \odot e_i^{mixup(k)} + e_i^{noise(k)}. \tag{15}$$

Similarly, we can derive the expression for $e_i^y$ using the same argument. The results in the lemma follow upon applying the rescaling $1 - \lambda \mapsto \epsilon(1-\lambda)$, $\sigma_{add} \mapsto \epsilon \sigma_{add}$ and $\sigma_{mult} \mapsto \epsilon \sigma_{mult}$, for $\epsilon > 0$. □

# B STATEMENTS AND PROOF OF THE RESULTS IN THE MAIN PAPER

## B.1 COMPLETE STATEMENT OF THEOREM 1 IN THE MAIN PAPER AND THE PROOF

We first state the complete statement of Theorem 1 in the main paper.

**Theorem 3** (Theorem 1 in the main paper). *Let $\epsilon > 0$ be a small parameter, and assume that $h$ and $f$ are twice differentiable. Then, $L_n^{NFM} = \mathbb{E}_{k \sim \mathcal{S}} L_n^{NFM(k)}$, where*

$$L_n^{NFM(k)} = L_n^{std} + \epsilon R_1^{(k)} + \epsilon^2 \tilde{R}_2^{(k)} + \epsilon^2 \tilde{R}_3^{(k)} + \epsilon^2 \varphi(\epsilon), \tag{16}$$

*with*

$$\tilde{R}_2^{(k)} = R_2^{(k)} + \sigma_{add}^2 R_2^{add(k)} + \sigma_{mult}^2 R_2^{mult(k)}, \tag{17}$$

$$\tilde{R}_3^{(k)} = R_3^{(k)} + \sigma_{add}^2 R_3^{add(k)} + \sigma_{mult}^2 R_3^{mult(k)}, \tag{18}$$

*where*

$$R_1^{(k)} = \frac{\mathbb{E}_{\lambda \sim \tilde{\mathcal{D}}_\lambda}[1-\lambda]}{n} \sum_{i=1}^n (h'(f(x_i) - y_i) \nabla_k f(g_k(x_i))^T \mathbb{E}_{x_r \sim \mathcal{D}_x}[g_k(x_r) - g_k(x_i)], \tag{19}$$

$$R_2^{(k)} = \frac{\mathbb{E}_{\lambda \sim \tilde{\mathcal{D}}_\lambda}[(1-\lambda)^2]}{2n} \sum_{i=1}^n h''(f(x_i)) \nabla_k f(g_k(x_i))^T$$
$$\times \mathbb{E}_{x_r \sim \mathcal{D}_x}[(g_k(x_r) - g_k(x_i))(g_k(x_r) - g_k(x_i))^T] \nabla_k f(g_k(x_i)), \tag{20}$$

$$R_3^{(k)} = \frac{\mathbb{E}_{\lambda \sim \tilde{\mathcal{D}}_\lambda}[(1-\lambda)^2]}{2n} \sum_{i=1}^n (h'(f(x_i)) - y_i)$$
$$\times \mathbb{E}_{x_r \sim \mathcal{D}_x}[(g_k(x_r) - g_k(x_i))^T \nabla_k^2 f(g_k(x_i))(g_k(x_r) - g_k(x_i))], \tag{21}$$

$$R_2^{add(k)} = \frac{1}{2n} \sum_{i=1}^n h''(f(x_i)) \nabla_k f(g_k(x_i))^T \mathbb{E}_{\boldsymbol{\xi}_k}[\xi_k^{add}(\xi_k^{add})^T] \nabla_k f(g_k(x_i)), \tag{22}$$

$$R_2^{mult(k)} = \frac{1}{2n} \sum_{i=1}^n h''(f(x_i)) \nabla_k f(g_k(x_i))^T (\mathbb{E}_{\boldsymbol{\xi}_k}[\xi_k^{mult}(\xi_k^{mult})^T] \odot g_k(x_i) g_k(x_i)^T) \nabla_k f(g_k(x_i)), \tag{23}$$

$$R_3^{add(k)} = \frac{1}{2n} \sum_{i=1}^n (h'(f(x_i)) - y_i) \mathbb{E}_{\boldsymbol{\xi}_k}[(\xi_k^{add})^T \nabla_k^2 f(g_k(x_i)) \xi_k^{add}], \tag{24}$$

$$R_3^{mult(k)} = \frac{1}{2n} \sum_{i=1}^n (h'(f(x_i)) - y_i) \mathbb{E}_{\boldsymbol{\xi}_k}[(\xi_k^{mult} \odot g_k(x_i))^T \nabla_k^2 f(g_k(x_i))(\xi_k^{mult} \odot g_k(x_i))], \tag{25}$$

*and* $\varphi(\epsilon) = \mathbb{E}_{\lambda \sim \tilde{\mathcal{D}}_\lambda} \mathbb{E}_{x_r \sim \mathcal{D}_x} \mathbb{E}_{\boldsymbol{\xi}_k \sim \mathcal{Q}}[\varphi(\epsilon)]$, *with* $\varphi$ *some function such that* $\lim_{\epsilon \to 0} \varphi(\epsilon) = 0$.

Following the setup of Zhang et al. (2020), we provide empirical results to show that the second order Taylor approximation for the NFM loss function is generally accurate (see Figure 7).

Recall from the main paper that the NFM loss function to be minimized is $L_n^{NFM} = \mathbb{E}_{k \sim \mathcal{S}} L_n^{NFM(k)}$, where

$$L_n^{NFM(k)} = \frac{1}{n^2} \sum_{i=1}^n \sum_{j=1}^n \mathbb{E}_{\lambda \sim Beta(\alpha,\beta)} \mathbb{E}_{\boldsymbol{\xi}_k \sim \mathcal{Q}} l(f_k(M_{\lambda,\boldsymbol{\xi}_k}(g_k(x_i), g_k(x_j))), M_\lambda(y_i, y_j)), \tag{26}$$

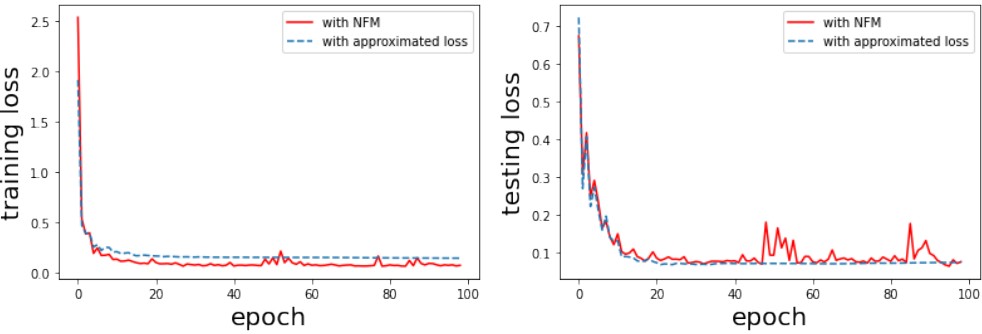

Figure 7: Comparison of the original NFM loss with the approximate loss function during training and testing for a two layer ReLU neural network trained on the toy dataset of Subsection F.2.

where $l : \mathbb{R}^K \times \mathbb{R}^K \to [0, \infty)$ is a loss function of the form $l(f(x), y) = h(f(x)) - yf(x)$, $\boldsymbol{\xi}_k := (\xi_k^{add}, \xi_k^{mult})$ are drawn from some probability distribution $\mathcal{Q}$ with finite first two moments (with zero mean), and

$$M_{\lambda, \boldsymbol{\xi}_k}(g_k(x), g_k(x')) := (\mathbb{1} + \sigma_{mult}\xi_k^{mult}) \odot M_\lambda(g_k(x), g_k(x')) + \sigma_{add}\xi_k^{add}. \qquad (27)$$

Before proving Theorem 3, we note that, following the argument of the proof of Lemma 3.1 in Zhang et al. (2020), the loss function minimized by NFM can be written as follows. For completeness, we provide all details of the proof.

**Lemma 2.** *The NFM loss (26) can be equivalently written as $L_n^{NFM} = \mathbb{E}_{k \sim \mathcal{S}} L_n^{NFM(k)}$, where*

$$L_n^{NFM(k)} = \frac{1}{n} \sum_{i=1}^n \mathbb{E}_{\lambda \sim \tilde{\mathcal{D}}_\lambda} \mathbb{E}_{x_r \sim \mathcal{D}_x} \mathbb{E}_{\boldsymbol{\xi}_k \sim \mathcal{Q}} [h(f_k(g_k(x_i) + \epsilon e_i^{NFM(k)})) - y_i f_k(g_k(x_i) + \epsilon e_i^{NFM(k)})], \tag{28}$$

*with*

$$e_i^{NFM(k)} = (\mathbb{1} + \epsilon \sigma_{mult} \xi_k^{mult}) \odot e_i^{mixup(k)} + e_i^{noise(k)}. \tag{29}$$

*Here $e_i^{mixup(k)} = (1 - \lambda)(g_k(x_r) - g_k(x_i))$ and $e_i^{noise(k)} = \sigma_{mult}\xi_k^{mult} \odot g_k(x_i) + \sigma_{add}\xi_k^{add}$, with $g_k(x_i), g_k(x_r) \in g_k(\mathcal{X})$ and $\lambda \sim Beta(\alpha, \beta)$.*

*Proof of Lemma 2.* From (26), we have:

$$L_n^{NFM(k)} = \frac{1}{n^2} \sum_{i=1}^n \sum_{j=1}^n \mathbb{E}_{\lambda \sim Beta(\alpha, \beta)} \mathbb{E}_{\boldsymbol{\xi}_k \sim \mathcal{Q}} l(f_k(M_{\lambda, \boldsymbol{\xi}_k}(g_k(x_i), g_k(x_j))), M_\lambda(y_i, y_j)). \tag{30}$$

We can rewrite:

$$\mathbb{E}_{\lambda \sim Beta(\alpha, \beta)} l(f_k(M_{\lambda, \boldsymbol{\xi}_k}(g_k(x_i), g_k(x_j))), M_\lambda(y_i, y_j))$$

$$= \mathbb{E}_{\lambda \sim Beta(\alpha, \beta)}[h(f_k(M_{\lambda, \boldsymbol{\xi}_k}(g_k(x_i), g_k(x_j)))) - M_\lambda(y_i, y_j) f_k(M_{\lambda, \boldsymbol{\xi}_k}(g_k(x_i), g_k(x_j)))] \tag{31}$$

$$= \mathbb{E}_{\lambda \sim Beta(\alpha, \beta)}[\lambda(h(f_k(M_{\lambda, \boldsymbol{\xi}_k}(g_k(x_i), g_k(x_j)))) - y_i f_k(M_{\lambda, \boldsymbol{\xi}_k}(g_k(x_i), g_k(x_j))))$$
$$+ (1 - \lambda)(h(f_k(M_{\lambda, \boldsymbol{\xi}_k}(g_k(x_i), g_k(x_j)))) - y_j f_k(M_{\lambda, \boldsymbol{\xi}_k}(g_k(x_i), g_k(x_j))))] \tag{32}$$

$$= \mathbb{E}_{\lambda \sim Beta(\alpha, \beta)} \mathbb{E}_{B \sim Bern(\lambda)}[B(h(f_k(M_{\lambda, \boldsymbol{\xi}_k}(g_k(x_i), g_k(x_j)))) - y_i f_k(M_{\lambda, \boldsymbol{\xi}_k}(g_k(x_i), g_k(x_j))))$$
$$+ (1 - B)(h(f_k(M_{\lambda, \boldsymbol{\xi}_k}(g_k(x_i), g_k(x_j)))) - y_j f_k(M_{\lambda, \boldsymbol{\xi}_k}(g_k(x_i), g_k(x_j))))], \tag{33}$$

where $Bern(\lambda)$ denotes the Bernoulli distribution with parameter $\lambda$ (i.e., $\mathbb{P}[B = 1] = \lambda$ and $\mathbb{P}[B = 0] = 1 - \lambda$).

Note that $\lambda \sim Beta(\alpha, \beta)$ and $B|\lambda \sim Bern(\lambda)$. By conjugacy, we can switch their order:

$$B \sim Bern\left(\frac{\alpha}{\alpha + \beta}\right), \quad \lambda|B \sim Beta(\alpha + B, \beta + 1 - B), \tag{34}$$

and arrive at:

$$\mathbb{E}_{\lambda \sim Beta(\alpha, \beta)} l(f_k(M_{\lambda, \boldsymbol{\xi}_k}(g_k(x_i), g_k(x_j))), M_\lambda(y_i, y_j))$$

$$= \mathbb{E}_{B \sim Bern\left(\frac{\alpha}{\alpha + \beta}\right)} \mathbb{E}_{\lambda \sim Beta(\alpha + B, \beta + 1 - B)}[B(h(f_k(M_{\lambda, \boldsymbol{\xi}_k}(g_k(x_i), g_k(x_j))))$$
$$- y_i f_k(M_{\lambda, \boldsymbol{\xi}_k}(g_k(x_i), g_k(x_j))))$$
$$+ (1 - B)(h(f_k(M_{\lambda, \boldsymbol{\xi}_k}(g_k(x_i), g_k(x_j)))) - y_j f_k(M_{\lambda, \boldsymbol{\xi}_k}(g_k(x_i), g_k(x_j))))] \tag{35}$$

$$= \frac{\alpha}{\alpha + \beta} \mathbb{E}_{\lambda \sim Beta(\alpha + 1, \beta)}[h(f_k(M_{\lambda, \boldsymbol{\xi}_k}(g_k(x_i), g_k(x_j)))) - y_i f_k(M_{\lambda, \boldsymbol{\xi}_k}(g_k(x_i), g_k(x_j)))]$$

$$+ \frac{\beta}{\alpha + \beta} \mathbb{E}_{\lambda \sim Beta(\alpha, \beta + 1)}[h(f_k(M_{\lambda, \boldsymbol{\xi}_k}(g_k(x_i), g_k(x_j)))) - y_j f_k(M_{\lambda, \boldsymbol{\xi}_k}(g_k(x_i), g_k(x_j)))]. \tag{36}$$

Using the facts that $Beta(\beta + 1, \alpha)$ and $1 - Beta(\alpha, \beta + 1)$ are of the same distribution and $M_{1-\lambda}(x_i, x_j) = M_\lambda(x_j, x_i)$, we have:

$$\sum_{i,j} \mathbb{E}_{\lambda \sim Beta(\alpha, \beta + 1)}[h(f_k(M_{\lambda, \boldsymbol{\xi}_k}(g_k(x_i), g_k(x_j)))) - y_j f_k(M_{\lambda, \boldsymbol{\xi}_k}(g_k(x_i), g_k(x_j)))]$$

$$= \sum_{i,j} \mathbb{E}_{\lambda \sim Beta(\beta + 1, \alpha)}[h(f_k(M_{\lambda, \boldsymbol{\xi}_k}(g_k(x_i), g_k(x_j)))) - y_i f_k(M_{\lambda, \boldsymbol{\xi}_k}(g_k(x_i), g_k(x_j)))]. \tag{37}$$

Therefore, denoting $\tilde{\mathcal{D}}_\lambda := \frac{\alpha}{\alpha+\beta} Beta(\alpha+1,\beta) + \frac{\beta}{\alpha+\beta} Beta(\beta+1,\alpha)$ and $\mathcal{D}_x := \frac{1}{n}\sum_{j=1}^n \delta_{x_j}$ the empirical distribution induced by the training samples $\{x_j\}_{j\in[n]}$, we have:

$$L_n^{NFM(k)} = \frac{1}{n}\sum_{i=1}^n \mathbb{E}_{\lambda\sim\tilde{\mathcal{D}}_\lambda}\mathbb{E}_{x_r\sim\mathcal{D}_x}\mathbb{E}_{\boldsymbol{\xi}_k\sim\mathcal{Q}}[h(f_k(M_{\lambda,\boldsymbol{\xi}_k}(g_k(x_i),g_k(x_r))))$$
$$- y_i f_k(M_{\lambda,\boldsymbol{\xi}_k}(g_k(x_i),g_k(x_r)))]. \tag{38}$$

The statement of the lemma follows upon substituting the fact that $M_{\lambda,\boldsymbol{\xi}_k}(g_k(x_i),g_k(x_r)) = g_k(x_i) + \epsilon e_i^{NFM(k)}$ into the above equation. $\qquad\square$

With this lemma in hand, we now prove Theorem 3.

*Proof of Theorem 3.* Denote $\psi_i(\epsilon) := h(f_k(g_k(x_i) + \epsilon e_i^{NFM(k)})) - y_i f_k(g_k(x_i) + \epsilon e_i^{NFM(k)})$, where $e_i^{NFM(k)}$ is given in (29). Since $h$ and $f_k$ are twice differentiable by assumption, $\psi_i$ is twice differentiable in $\epsilon$, and

$$\psi_i(\epsilon) = \psi_i(0) + \epsilon\psi_i'(0) + \frac{\epsilon^2}{2}\psi_i''(0) + \epsilon^2\varphi_i(\epsilon), \tag{39}$$

where $\varphi_i$ is some function such that $\lim_{\epsilon\to 0}\varphi_i(\epsilon) = 0$. Therefore, by Lemma 2, $L_n^{NFM} = \mathbb{E}_{k\sim\mathcal{S}}L_n^{NFM(k)}$, where

$$L_n^{NFM(k)} = \frac{1}{n}\sum_{i=1}^n \mathbb{E}_{\lambda\sim\tilde{\mathcal{D}}_\lambda}\mathbb{E}_{x_r\sim\mathcal{D}_x}\mathbb{E}_{\boldsymbol{\xi}_k\sim\mathcal{Q}}[\psi_i(\epsilon)] \tag{40}$$

$$= \frac{1}{n}\sum_{i=1}^n \mathbb{E}_{\lambda\sim\tilde{\mathcal{D}}_\lambda}\mathbb{E}_{x_r\sim\mathcal{D}_x}\mathbb{E}_{\boldsymbol{\xi}_k\sim\mathcal{Q}}\left[\psi_i(0) + \epsilon\psi_i'(0) + \frac{\epsilon^2}{2}\psi_i''(0) + \epsilon^2\varphi_i(\epsilon)\right] \tag{41}$$

$$= \frac{1}{n}\sum_{i=1}^n \mathbb{E}_{\lambda\sim\tilde{\mathcal{D}}_\lambda}\mathbb{E}_{x_r\sim\mathcal{D}_x}\mathbb{E}_{\boldsymbol{\xi}_k\sim\mathcal{Q}}\left[\psi_i(0) + \epsilon\psi_i'(0) + \frac{\epsilon^2}{2}\psi_i''(0)\right] + \epsilon^2\varphi(\epsilon) \tag{42}$$

$$=: L_n^{std} + \epsilon R_1^{(k)} + \epsilon^2(\tilde{R}_2^{(k)} + \tilde{R}_3^{(k)}) + \epsilon^2\varphi(\epsilon), \tag{43}$$

where $\varphi(\epsilon) = \frac{1}{n}\sum_{i=1}^n \mathbb{E}_{\lambda\sim\tilde{\mathcal{D}}_\lambda}\mathbb{E}_{x_r\sim\mathcal{D}_x}\mathbb{E}_{\boldsymbol{\xi}_k\sim\mathcal{Q}}[\varphi_i(\epsilon)]$.

It remains to compute $\psi_i'(0)$ and $\psi_i''(0)$ in order to arrive at the expression for the $R_1^{(k)}$, $\tilde{R}_2^{(k)}$ and $\tilde{R}_3^{(k)}$ presented in Theorem 3.

Denoting $\tilde{g}_k(x_i) := g_k(x_i) + \epsilon e_i^{NFM(k)}$, we compute, applying chain rule:

$$\psi_i'(\epsilon) = h'(f_k(\tilde{g}_k(x_i)))\nabla_k f_k(\tilde{g}_k(x_i))^T\frac{\partial\tilde{g}_k(x_i)}{\partial\epsilon} - y_i\nabla_k f_k(\tilde{g}_k(x_i))^T\frac{\partial\tilde{g}_k(x_i)}{\partial\epsilon} \tag{44}$$

$$= (h'(f_k(\tilde{g}_k(x_i))) - y_i)\nabla_k f_k(\tilde{g}_k(x_i))^T\frac{\partial\tilde{g}_k(x_i)}{\partial\epsilon} \tag{45}$$

$$= (h'(f_k(\tilde{g}_k(x_i))) - y_i)\nabla_k f_k(\tilde{g}_k(x_i))^T e_i^{NFM(k)} \tag{46}$$

$$= (h'(f_k(\tilde{g}_k(x_i))) - y_i)\nabla_k f_k(\tilde{g}_k(x_i))^T[(1-\lambda)(g_k(x_r) - g_k(x_i)) + \sigma_{add}\xi_k^{add}$$
$$+ \sigma_{mult}\xi_k^{mult}\odot g_k(x_i) + \epsilon(1-\lambda)\sigma_{mult}\xi_k^{mult}\odot(g_k(x_r) - g_k(x_i))], \tag{47}$$

where we have used $\frac{\partial\tilde{g}_k(x_i)}{\partial\epsilon} = e_i^{NFM(k)}$ in the second last line and substituted the expression for $e_i^{NFM(k)}$ from (29) in the last line above.

Therefore,

$$\psi_i'(0) = (h'(f_k(g_k(x_i))) - y_i)\nabla_k f_k(g_k(x_i))^T[(1-\lambda)(g_k(x_r) - g_k(x_i)) + \sigma_{add}\xi_k^{add}$$
$$+ \sigma_{mult}\xi_k^{mult}\odot g_k(x_i)], \tag{48}$$

and

$$\mathbb{E}_{\boldsymbol{\xi}_k \sim \mathcal{Q}} \psi_i'(0) = (h'(f_k(g_k(x_i))) - y_i) \nabla_k f_k(g_k(x_i))^T [(1 - \lambda)(g_k(x_r) - g_k(x_i))], \quad (49)$$

where we have used the assumptions that $\mathbb{E}_{\boldsymbol{\xi}_k \sim \mathcal{Q}} \xi_k^{add} = 0$ and $\mathbb{E}_{\boldsymbol{\xi}_k \sim \mathcal{Q}} \xi_k^{mult} = 0$. The expression for the $R_1^{(k)}$ in the theorem then follows from substituting (49) into (42).

Next, using chain rule, we have:

$$\psi_i''(\epsilon) = \frac{\partial}{\partial \epsilon} \left( (h'(f_k(\tilde{g}_k(x_i))) - y_i) \nabla_k f_k(\tilde{g}_k(x_i))^T \frac{\partial \tilde{g}_k(x_i)}{\partial \epsilon} \right) \quad (50)$$

$$= \left( \frac{\partial}{\partial \epsilon} (h'(f_k(\tilde{g}_k(x_i))) - y_i) \right) \nabla_k f_k(\tilde{g}_k(x_i))^T \frac{\partial \tilde{g}_k(x_i)}{\partial \epsilon}$$

$$+ (h'(f_k(\tilde{g}_k(x_i))) - y_i) \frac{\partial}{\partial \epsilon} \left( \nabla_k f_k(\tilde{g}_k(x_i))^T \frac{\partial \tilde{g}_k(x_i)}{\partial \epsilon} \right). \quad (51)$$

Note that, applying chain rule,

$$\frac{\partial}{\partial \epsilon} \left( \nabla_k f_k(\tilde{g}_k(x_i))^T \frac{\partial \tilde{g}_k(x_i)}{\partial \epsilon} \right) = \frac{\partial}{\partial \epsilon} \left( \nabla_k f_k(\tilde{g}_k(x_i))^T e_i^{NFM(k)} \right) \quad (52)$$

$$= \frac{\partial}{\partial \epsilon} \left( (e_i^{NFM(k)})^T \nabla_k f_k(\tilde{g}_k(x_i)) \right) \quad (53)$$

$$= (e_i^{NFM(k)})^T \nabla_k^2 f_k(\tilde{g}_k(x_i)) \frac{\partial \tilde{g}_k(x_i)}{\partial \epsilon} \quad (54)$$

$$= (e_i^{NFM(k)})^T \nabla_k^2 f_k(\tilde{g}_k(x_i)) e_i^{NFM(k)}. \quad (55)$$

Also, using chain rule again,

$$\left( \frac{\partial}{\partial \epsilon} (h'(f_k(\tilde{g}_k(x_i))) - y_i) \right) = h''(f_k(\tilde{g}_k(x_i))) \nabla_k f_k(\tilde{g}_k(x_i))^T \frac{\partial \tilde{g}_k(x_i)}{\partial \epsilon} \quad (56)$$

$$= h''(f_k(\tilde{g}_k(x_i))) \nabla_k f_k(\tilde{g}_k(x_i))^T e_i^{NFM(k)}. \quad (57)$$

Therefore, we have:

$$\psi_i''(\epsilon) = h''(f_k(\tilde{g}_k(x_i))) \nabla_k f_k(\tilde{g}_k(x_i))^T e_i^{NFM(k)} (e_i^{NFM(k)})^T \nabla_k f_k(\tilde{g}_k(x_i))$$
$$+ (h'(f_k(\tilde{g}_k(x_i))) - y_i)(e_i^{NFM(k)})^T \nabla_k^2 f_k(\tilde{g}_k(x_i)) e_i^{NFM(k)} \quad (58)$$
$$= h''(f_k(\tilde{g}_k(x_i))) \nabla_k f_k(\tilde{g}_k(x_i))^T [(1 - \lambda)(g_k(x_r) - g_k(x_i)) + \sigma_{add} \xi_k^{add}$$
$$+ \sigma_{mult} \xi_k^{mult} \odot g_k(x_i) + \epsilon(1 - \lambda) \sigma_{mult} \xi_k^{mult} \odot (g_k(x_r) - g_k(x_i))]$$
$$\times [(1 - \lambda)(g_k(x_r) - g_k(x_i)) + \sigma_{add} \xi_k^{add} + \sigma_{mult} \xi_k^{mult} \odot g_k(x_i)$$
$$+ \epsilon(1 - \lambda) \sigma_{mult} \xi_k^{mult} \odot (g_k(x_r) - g_k(x_i))]^T \nabla_k f_k(\tilde{g}_k(x_i))$$
$$+ (h'(f_k(\tilde{g}_k(x_i))) - y_i)[(1 - \lambda)(g_k(x_r) - g_k(x_i)) + \sigma_{add} \xi_k^{add} + \sigma_{mult} \xi_k^{mult} \odot g_k(x_i)$$
$$+ \epsilon(1 - \lambda) \sigma_{mult} \xi_k^{mult} \odot (g_k(x_r) - g_k(x_i))]^T \nabla_k^2 f_k(\tilde{g}_k(x_i))[(1 - \lambda)(g_k(x_r) - g_k(x_i))$$
$$+ \sigma_{add} \xi_k^{add} + \sigma_{mult} \xi_k^{mult} \odot g_k(x_i) + \epsilon(1 - \lambda) \sigma_{mult} \xi_k^{mult} \odot (g_k(x_r) - g_k(x_i))] \quad (59)$$
$$=: h''(f_k(\tilde{g}_k(x_i))) \nabla_k f_k(\tilde{g}_k(x_i))^T P_1(\epsilon) \nabla_k f_k(\tilde{g}_k(x_i)) + (h'(f_k(\tilde{g}_k(x_i))) - y_i) P_2(\epsilon),$$
$$(60)$$

where we have substituted the expression for the $e_i^{NFM(k)}$ into the first line to arrive at the last line above.

Note that,

$$
\begin{aligned}
&\mathbb{E}_{\boldsymbol{\xi}_k \sim \mathcal{Q}} P_1(\epsilon) \\
&= \mathbb{E}_{\boldsymbol{\xi}_k \sim \mathcal{Q}}[(1-\lambda)(g_k(x_r) - g_k(x_i)) + \sigma_{add}\xi_k^{add} + \sigma_{mult}\xi_k^{mult} \odot g_k(x_i) \\
&\quad + \epsilon(1-\lambda)\sigma_{mult}\xi_k^{mult} \odot (g_k(x_r) - g_k(x_i))] \times [(1-\lambda)(g_k(x_r) - g_k(x_i)) + \sigma_{add}\xi_k^{add} \\
&\quad + \sigma_{mult}\xi_k^{mult} \odot g_k(x_i) + \epsilon(1-\lambda)\sigma_{mult}\xi_k^{mult} \odot (g_k(x_r) - g_k(x_i))]^T \quad (61) \\
&= (1-\lambda)^2(g_k(x_r) - g_k(x_i))(g_k(x_r) - g_k(x_i))^T + \sigma_{add}^2 \mathbb{E}_{\boldsymbol{\xi}_k \sim \mathcal{Q}}[\xi_k^{add}(\xi_k^{add})^T] \\
&\quad + \sigma_{mult}^2 \mathbb{E}_{\boldsymbol{\xi}_k \sim \mathcal{Q}}[(\xi_k^{mult} \odot g_k(x_i))(\xi_k^{mult} \odot g_k(x_i))^T] + o(\epsilon) \quad (62) \\
&= (1-\lambda)^2(g_k(x_r) - g_k(x_i))(g_k(x_r) - g_k(x_i))^T + \sigma_{add}^2 \mathbb{E}_{\boldsymbol{\xi}_k \sim \mathcal{Q}}[\xi_k^{add}(\xi_k^{add})^T] \\
&\quad + \sigma_{mult}^2 \mathbb{E}_{\boldsymbol{\xi}_k \sim \mathcal{Q}}[(\xi_k^{mult}(\xi_k^{mult})^T) \odot g_k(x_i))g_k(x_i)^T] + o(\epsilon), \quad (63)
\end{aligned}
$$

as $\epsilon \to 0$, where we have used the assumption that $\mathbb{E}_{\boldsymbol{\xi}_k \sim \mathcal{Q}}\xi_k^{add} = 0$ and $\mathbb{E}_{\boldsymbol{\xi}_k \sim \mathcal{Q}}\xi_k^{mult} = 0$ in the second last line above.

Similarly,

$$
\begin{aligned}
&\mathbb{E}_{\boldsymbol{\xi}_k \sim \mathcal{Q}} P_2(\epsilon) \\
&= \mathbb{E}_{\boldsymbol{\xi}_k \sim \mathcal{Q}}[(1-\lambda)(g_k(x_r) - g_k(x_i)) + \sigma_{add}\xi_k^{add} + \sigma_{mult}\xi_k^{mult} \odot g_k(x_i) \\
&\quad + \epsilon(1-\lambda)\sigma_{mult}\xi_k^{mult} \odot (g_k(x_r) - g_k(x_i))]^T \nabla_k^2 f_k(\tilde{g}_k(x_i))[(1-\lambda)(g_k(x_r) - g_k(x_i)) \\
&\quad + \sigma_{add}\xi_k^{add} + \sigma_{mult}\xi_k^{mult} \odot g_k(x_i) + \epsilon(1-\lambda)\sigma_{mult}\xi_k^{mult} \odot (g_k(x_r) - g_k(x_i))] \quad (64) \\
&= (1-\lambda)^2(g_k(x_r) - g_k(x_i))^T \nabla_k^2 f_k(\tilde{g}_k(x_i))(g_k(x_r) - g_k(x_i)) \\
&\quad + \sigma_{add}^2 \mathbb{E}_{\boldsymbol{\xi}_k \sim \mathcal{Q}}[(\xi_k^{add})^T \nabla_k^2 f_k(\tilde{g}_k(x_i))\xi_k^{add}] \\
&\quad + \sigma_{mult}^2 \mathbb{E}_{\boldsymbol{\xi}_k \sim \mathcal{Q}}[(\xi_k^{mult} \odot g_k(x_i))^T \nabla_k^2 f_k(\tilde{g}_k(x_i))(\xi_k^{mult} \odot g_k(x_i))] + o(\epsilon), \quad (65)
\end{aligned}
$$

as $\epsilon \to 0$.

Now, recall from Eq. (42) that we have

$$
L_n^{NFM(k)} = \frac{1}{n}\sum_{i=1}^n \mathbb{E}_{\lambda \sim \tilde{\mathcal{D}}_\lambda} \mathbb{E}_{x_r \sim \mathcal{D}_x} \mathbb{E}_{\boldsymbol{\xi}_k \sim \mathcal{Q}} \left[ \psi_i(0) + \epsilon\psi_i'(0) + \frac{\epsilon^2}{2}\psi_i''(0) \right] + \epsilon^2 \varphi(\epsilon) \quad (66)
$$

$$
=: L_n^{std} + \epsilon R_1^{(k)} + \epsilon^2(\tilde{R}_2^{(k)} + \tilde{R}_3^{(k)}) + \epsilon^2 \varphi(\epsilon), \quad (67)
$$

where $\psi_i(0) = h(f_k(g_k(x_i))) - y_i f_k(g_k(x_i))$. Also, we have:

$$
\begin{aligned}
&\mathbb{E}_{\boldsymbol{\xi}_k \sim \mathcal{Q}}[\psi_i''(\epsilon)] \\
&= h''(f_k(\tilde{g}_k(x_i)))\nabla_k f_k(\tilde{g}_k(x_i))^T \mathbb{E}_{\boldsymbol{\xi}_k \sim \mathcal{Q}}[P_1(\epsilon)]\nabla_k f_k(\tilde{g}_k(x_i)) \\
&\quad + (h'(f_k(\tilde{g}_k(x_i))) - y_i)\mathbb{E}_{\boldsymbol{\xi}_k \sim \mathcal{Q}}[P_2(\epsilon)] \quad (68) \\
&= h''(f_k(\tilde{g}_k(x_i)))\nabla_k f_k(\tilde{g}_k(x_i))^T[(1-\lambda)^2(g_k(x_r) - g_k(x_i))(g_k(x_r) - g_k(x_i))^T \\
&\quad + \sigma_{add}^2 \mathbb{E}_{\boldsymbol{\xi}_k \sim \mathcal{Q}}[\xi_k^{add}(\xi_k^{add})^T] + \sigma_{mult}^2 \mathbb{E}_{\boldsymbol{\xi}_k \sim \mathcal{Q}}[(\xi_k^{mult}(\xi_k^{mult})^T) \odot g_k(x_i))g_k(x_i)^T] + o(\epsilon)] \\
&\quad \times \nabla_k f_k(\tilde{g}_k(x_i)) \\
&\quad + (h'(f_k(\tilde{g}_k(x_i))) - y_i)[(1-\lambda)^2(g_k(x_r) - g_k(x_i))^T \nabla_k^2 f_k(\tilde{g}_k(x_i))(g_k(x_r) - g_k(x_i)) \\
&\quad + \sigma_{add}^2 \mathbb{E}_{\boldsymbol{\xi}_k \sim \mathcal{Q}}[(\xi_k^{add})^T \nabla_k^2 f_k(\tilde{g}_k(x_i))\xi_k^{add}] \\
&\quad + \sigma_{mult}^2 \mathbb{E}_{\boldsymbol{\xi}_k \sim \mathcal{Q}}[(\xi_k^{mult} \odot g_k(x_i))^T \nabla_k^2 f_k(\tilde{g}_k(x_i))(\xi_k^{mult} \odot g_k(x_i))] + o(\epsilon)]. \quad (69)
\end{aligned}
$$

Therefore, setting $\epsilon = 0$,

$$
\begin{aligned}
&\mathbb{E}_{\boldsymbol{\xi}_k \sim \mathcal{Q}}[\psi_i''(0)] \\
&= h''(f_k(g_k(x_i))) \nabla_k f_k(g_k(x_i))^T [(1-\lambda)^2 (g_k(x_r) - g_k(x_i))(g_k(x_r) - g_k(x_i))^T \\
&\quad + \sigma_{add}^2 \mathbb{E}_{\boldsymbol{\xi}_k \sim \mathcal{Q}}[\xi_k^{add}(\xi_k^{add})^T] + \sigma_{mult}^2 \mathbb{E}_{\boldsymbol{\xi}_k \sim \mathcal{Q}}[(\xi_k^{mult}(\xi_k^{mult})^T) \odot g_k(x_i))g_k(x_i)^T]] \\
&\quad \times \nabla_k f_k(g_k(x_i)) \\
&\quad + (h'(f_k(g_k(x_i))) - y_i)[(1-\lambda)^2 (g_k(x_r) - g_k(x_i))^T \nabla_k^2 f_k(g_k(x_i))(g_k(x_r) - g_k(x_i)) \\
&\quad + \sigma_{add}^2 \mathbb{E}_{\boldsymbol{\xi}_k \sim \mathcal{Q}}[(\xi_k^{add})^T \nabla_k^2 f_k(g_k(x_i)) \xi_k^{add}] \\
&\quad + \sigma_{mult}^2 \mathbb{E}_{\boldsymbol{\xi}_k \sim \mathcal{Q}}[(\xi_k^{mult} \odot g_k(x_i))^T \nabla_k^2 f_k(g_k(x_i))(\xi_k^{mult} \odot g_k(x_i))]].
\end{aligned}
\tag{70}
$$

The expression for the $\tilde{R}_2^{(k)}$ and $\tilde{R}_3^{(k)}$ in the theorem follows upon substituting (70) into (66). $\qquad \square$

## B.2 THEOREM 2 IN THE MAIN PAPER AND THE PROOF

We first restate Theorem 2 in the main paper and then provide the proof. Recall that we consider the binary cross-entropy loss, setting $h(z) = \log(1 + e^z)$, with the labels $y$ taking value in $\{0, 1\}$ and the classifier model $f : \mathbb{R}^d \to \mathbb{R}$.

**Theorem 4** (Theorem 2 in the main paper). *Let* $\theta \in \Theta := \{\theta : y_i f(x_i) + (y_i - 1) f(x_i) \geq 0 \text{ for all } i \in [n]\}$ *be a point such that* $\nabla_k f(g_k(x_i))$ *and* $\nabla_k^2 f(g_k(x_i))$ *exist for all* $i \in [n]$, $k \in \mathcal{S}$. *Assume that* $f_k(g_k(x_i)) = \nabla_k f(g_k(x_i))^T g_k(x_i)$, $\nabla_k^2 f(g_k(x_i)) = 0$ *for all* $i \in [n]$, $k \in \mathcal{S}$. *In addition, suppose that* $\|\nabla f(x_i)\|_2 > 0$ *for all* $i \in [n]$, $\mathbb{E}_{r \sim \mathcal{D}_x}[g_k(r)] = 0$ *and* $\|g_k(x_i)\|_2 \geq c_x^{(k)} \sqrt{d_k}$ *for all* $i \in [n]$, $k \in \mathcal{S}$. *Then,*

$$
L_n^{NFM} \geq \frac{1}{n} \sum_{i=1}^n \max_{\|\delta_i\|_2 \leq \epsilon_i^{mix}} l(f(x_i + \delta_i), y_i) + L_n^{reg} + \epsilon^2 \phi(\epsilon),
\tag{71}
$$

*where*

$$
\epsilon_i^{mix} = \epsilon \mathbb{E}_{\lambda \sim \tilde{\mathcal{D}}_\lambda}[1 - \lambda] \cdot \mathbb{E}_{k \sim \mathcal{S}} \left[ r_i^{(k)} c_x^{(k)} \frac{\|\nabla_k f(g_k(x_i))\|_2}{\|\nabla f(x_i)\|_2} \sqrt{d_k} \right],
\tag{72}
$$

$$
r_i^{(k)} = |\cos(\nabla_k f(g_k(x_i)), g_k(x_i))|,
\tag{73}
$$

$$
L_n^{reg} = \frac{1}{2n} \sum_{i=1}^n |h''(f(x_i))| (\epsilon_i^{reg})^2,
\tag{74}
$$

*with*

$$
\begin{aligned}
(\epsilon_i^{reg})^2 = \epsilon^2 \|\nabla_k f(g_k(x_i))\|_2^2 \Big( &\mathbb{E}_\lambda[(1-\lambda)]^2 \mathbb{E}_{x_r}[\|g_k(x_r)\|_2^2 \cos(\nabla_k f(g_k(x_i)), g_k(x_r))^2] \\
&+ \sigma_{add}^2 \mathbb{E}_{\boldsymbol{\xi}}[\|\xi_{add}\|_2^2 \cos(\nabla_k f(g_k(x_i)), \xi_{add})^2] \\
&+ \sigma_{mult}^2 \mathbb{E}_{\boldsymbol{\xi}}[\|\xi_{mult} \odot g_k(x_i)\|_2^2 \cos(\nabla_k f(g_k(x_i)), \xi_{mult} \odot g_k(x_i))^2] \Big),
\end{aligned}
\tag{75}
$$

*and* $\phi$ *is some function such that* $\lim_{\epsilon \to 0} \phi(\epsilon) = 0$.

Theorem 4 says that $L_n^{NFM}$ is approximately an upper bound of sum of an adversarial loss with $l_2$-attack of size $\epsilon^{mix} = \min_i \epsilon_i^{mix}$ and a feature-dependent regularizer with the strength of $\min_i (\epsilon_i^{reg})^2$. Therefore, minimizing the NFM loss would result in a small regularized adversarial loss. We note that both $\epsilon_i^{mix}$ and $\epsilon_i^{reg}$ depend on the cosine similarities between the directional derivatives and the features at which the derivatives are evaluated at, whereas the $\epsilon_i^{reg}$ additionally depend on the cosine similarities between the directional derivatives and the injected noise.

Before proving Theorem 4, we remark that the assumption that $f_k(g_k(x_i)) = \nabla_k f(g_k(x_i))^T g_k(x_i)$, $\nabla_k^2 f(g_k(x_i)) = 0$ for all $i \in [n]$, $k \in \mathcal{S}$ is satisfied by fully connected neural networks with ReLU activation function or max-pooling. For a proof of this, we refer to Section B.2 in Zhang et al. (2020). The assumption that $\mathbb{E}_{r \sim \mathcal{D}_x}[g_k(r)] = 0$ could be relaxed at the cost of obtaining a more complicated formula (see Remark 1 for the formula) for the $\epsilon_i^{reg}$ in the bound, which could be derived in a straightforward manner.

*Proof of Theorem 4.* For $h(z) = \log(1 + e^z)$, we have $h'(z) = \frac{e^z}{1+e^z} =: S(z) \geq 0$ and $h''(z) = \frac{e^z}{(1+e^z)^2} = S(z)(1 - S(z)) \geq 0$. Substituting these expressions into the equation of Theorem 3 and using the assumptions that $f_k(g_k(x_i)) = \nabla_k f(g_k(x_i))^T g_k(x_i)$ and $\mathbb{E}_{r \sim \mathcal{D}_x}[g_k(r)] = 0$, we have, for $k \in \mathcal{S}$,

$$R_1^{(k)} = \frac{\mathbb{E}_{\lambda \sim \tilde{\mathcal{D}}_\lambda}[1 - \lambda]}{n} \sum_{i=1}^n (y_i - S(f(x_i))) f_k(g_k(x_i)), \tag{76}$$

and we compute:

$$R_2^{(k)} = \frac{\mathbb{E}_{\lambda \sim \tilde{\mathcal{D}}_\lambda}[(1 - \lambda)^2]}{2n} \sum_{i=1}^n S(f(x_i))(1 - S(f(x_i))) \nabla_k f(g_k(x_i))^T$$
$$\times \mathbb{E}_{x_r \sim \mathcal{D}_x}[(g_k(x_r) - g_k(x_i))(g_k(x_r) - g_k(x_i))^T] \nabla_k f(g_k(x_i)) \tag{77}$$

$$\geq \frac{\mathbb{E}_{\lambda \sim \tilde{\mathcal{D}}_\lambda}[(1 - \lambda)]^2}{2n} \sum_{i=1}^n |S(f(x_i))(1 - S(f(x_i)))| \nabla_k f(g_k(x_i))^T$$
$$\times \mathbb{E}_{x_r \sim \mathcal{D}_x}[(g_k(x_r) - g_k(x_i))(g_k(x_r) - g_k(x_i))^T] \nabla_k f(g_k(x_i)) \tag{78}$$

$$= \frac{\mathbb{E}_{\lambda \sim \tilde{\mathcal{D}}_\lambda}[(1 - \lambda)]^2}{2n} \sum_{i=1}^n |S(f(x_i))(1 - S(f(x_i)))| \nabla_k f(g_k(x_i))^T$$
$$\times (\mathbb{E}_{x_r \sim \mathcal{D}_x}[(g_k(x_r) g_k(x_r)^T] + g_k(x_i) g_k(x_i)^T]) \nabla_k f(g_k(x_i)) \tag{79}$$

$$= \frac{\mathbb{E}_{\lambda \sim \tilde{\mathcal{D}}_\lambda}[(1 - \lambda)]^2}{2n} \sum_{i=1}^n |S(f(x_i))(1 - S(f(x_i)))| (\nabla_k f(g_k(x_i))^T g_k(x_i))^2$$
$$+ \frac{\mathbb{E}_{\lambda \sim \tilde{\mathcal{D}}_\lambda}[(1 - \lambda)]^2}{2n} \sum_{i=1}^n |S(f(x_i))(1 - S(f(x_i)))| \mathbb{E}_{x_r \in \mathcal{D}_x}[(\nabla_k f(g_k(x_i))^T g_k(x_r))^2] \tag{80}$$

$$= \frac{\mathbb{E}_{\lambda \sim \tilde{\mathcal{D}}_\lambda}[(1 - \lambda)]^2}{2n} \sum_{i=1}^n |S(f(x_i))(1 - S(f(x_i)))| \|\nabla_k f(g_k(x_i))\|_2^2 \|g_k(x_i)\|_2^2$$
$$\times (\cos(\nabla_k f(g_k(x_i)), g_k(x_i)))^2 + \frac{1}{2n} \sum_{i=1}^n |S(f(x_i))(1 - S(f(x_i)))| \|\nabla_k f(g_k(x_i))\|_2^2$$
$$\times \mathbb{E}_\lambda[(1 - \lambda)]^2 \mathbb{E}_{x_r}[\|g_k(x_r)\|_2^2 \cos(\nabla_k f(g_k(x_i)), g_k(x_r))^2] \tag{81}$$

$$\geq \frac{1}{2n} \sum_{i=1}^n |S(f(x_i))(1 - S(f(x_i)))| \|\nabla_k f(g_k(x_i))\|_2^2 \mathbb{E}_{\lambda \sim \tilde{\mathcal{D}}_\lambda}[(1 - \lambda)]^2 d_k(r_i^{(k)} c_x^{(k)})^2$$
$$+ \frac{1}{2n} \sum_{i=1}^n |S(f(x_i))(1 - S(f(x_i)))| \|\nabla_k f(g_k(x_i))\|_2^2 \cdot \mathbb{E}_\lambda[(1 - \lambda)]^2 \mathbb{E}_{x_r}[\|g_k(x_r)\|_2^2$$
$$\times \cos(\nabla_k f(g_k(x_i)), g_k(x_r))^2] \tag{82}$$

$$= \frac{1}{2n} \sum_{i=1}^n |S(f(x_i))(1 - S(f(x_i)))| \|\nabla f(x_i)\|_2^2$$
$$\times \left( \mathbb{E}_{\lambda \sim \tilde{\mathcal{D}}_\lambda}[(1 - \lambda)]^2 \frac{\|\nabla_k f(g_k(x_i))\|_2^2}{\|\nabla f(x_i)\|_2^2} d_k(r_i^{(k)} c_x^{(k)})^2 \right)$$
$$+ \frac{1}{2n} \sum_{i=1}^n |S(f(x_i))(1 - S(f(x_i)))| \|\nabla_k f(g_k(x_i))\|_2^2 \cdot \mathbb{E}_\lambda[(1 - \lambda)]^2 \mathbb{E}_{x_r}[\|g_k(x_r)\|_2^2$$
$$\times \cos(\nabla_k f(g_k(x_i)), g_k(x_r))^2]. \tag{83}$$

In the above, we have used the facts that $\mathbb{E}[Z^2] = \mathbb{E}[Z]^2 + Var(Z) \geq \mathbb{E}[Z]^2$ and $S, S(1 - S) \geq 0$ to obtain (78), the assumption that $\mathbb{E}_{r \sim \mathcal{D}_x}[g_k(r)] = 0$ to arrive at (79), the assumption that $\|g_k(x_i)\|_2 \geq c_x^{(k)} \sqrt{d_k}$ for all $i \in [n]$, $k \in \mathcal{S}$ to arrive at (82), and the assumption that $\|\nabla f(x_i)\|_2 > 0$ for all $i \in [n]$ to justify the last equation above.

Next, we bound $R_1^{(k)}$, using the assumption that $\theta \in \Theta$. Note that from our assumption on $\theta$, we have $y_i f(x_i) + (y_i - 1)f(x_i) \geq 0$, which implies that $f(x_i) \geq 0$ if $y_i = 1$ and $f(x_i) \leq 0$ if $y_i = 0$. Thus, if $y_i = 1$, then $(y_i - S(f(x_i)))f_k(g_k(x_i)) = (1 - S(f(x_i)))f_k(g_k(x_i)) \geq 0$, since $f(x_i) \geq 0$ and $(1 - S(f(x_i))) \geq 0$ due to the fact that $S(f(x_i)) \in (0, 1)$. A similar argument leads to $(y_i - S(f(x_i)))f_k(g_k(x_i)) \geq 0$ if $y_i = 0$. So, we have $(y_i - S(f(x_i)))f_k(g_k(x_i)) \geq 0$ for all $i \in [n]$.

Therefore, noting that $\mathbb{E}_{\lambda \sim \tilde{\mathcal{D}}_\lambda}[1 - \lambda] \geq 0$, we compute:

$$R_1^{(k)} = \frac{\mathbb{E}_{\lambda \sim \tilde{\mathcal{D}}_\lambda}[1 - \lambda]}{n} \sum_{i=1}^n |y_i - S(f(x_i))||f_k(g_k(x_i))| \tag{84}$$

$$= \frac{\mathbb{E}_{\lambda \sim \tilde{\mathcal{D}}_\lambda}[1 - \lambda]}{n} \sum_{i=1}^n |S(f(x_i)) - y_i| \|\nabla_k f(g_k(x_i))\|_2 \|g_k(x_i)\|_2 |\cos(\nabla_k f(g_k(x_i)), g_k(x_i))| \tag{85}$$

$$\geq \frac{1}{n} \sum_{i=1}^n |S(f(x_i)) - y_i| \|\nabla_k f(g_k(x_i))\|_2 (\mathbb{E}_{\lambda \sim \tilde{\mathcal{D}}_\lambda}[1 - \lambda] r_i^{(k)} c_x^{(k)} \sqrt{d_k}) \tag{86}$$

$$= \frac{1}{n} \sum_{i=1}^n |S(f(x_i)) - y_i| \|\nabla f(x_i)\|_2 \left( \mathbb{E}_{\lambda \sim \tilde{\mathcal{D}}_\lambda}[1 - \lambda] \frac{\|\nabla_k f(g_k(x_i))\|_2}{\|\nabla f(x_i)\|_2} r_i^{(k)} c_x^{(k)} \sqrt{d_k} \right). \tag{87}$$

Note that $R_3^{(k)} = 0$ as a consequence of our assumption that $\nabla_k^2 f(g_k(x_i)) = 0$ for all $i \in [n]$, $k \in \mathcal{S}$, and similar argument leads to:

$$R_2^{add(k)} = \frac{1}{2n} \sum_{i=1}^n |S(f(x_i))(1 - S(f(x_i)))| \nabla_k f(g_k(x_i))^T \mathbb{E}_{\boldsymbol{\xi}_k}[\xi_k^{add}(\xi_k^{add})^T] \nabla_k f(g_k(x_i)) \tag{88}$$

$$= \frac{1}{2n} \sum_{i=1}^n |S(f(x_i))(1 - S(f(x_i)))| \|\nabla_k f(g_k(x_i))\|_2^2$$
$$\times \mathbb{E}_{\boldsymbol{\xi}_k}[\|\xi_k^{add}\|_2^2 \cos(\nabla_k f(g_k(x_i)), \xi_k^{add})^2] \tag{89}$$

$$R_2^{mult(k)} = \frac{1}{2n} \sum_{i=1}^n |S(f(x_i))(1 - S(f(x_i)))| \nabla_k f(g_k(x_i))^T (\mathbb{E}_{\boldsymbol{\xi}_k}[\xi_k^{add}(\xi_k^{add})^T] \odot g_k(x_i)g_k(x_i)^T)$$
$$\times \nabla_k f(g_k(x_i))$$
$$= \frac{1}{2n} \sum_{i=1}^n |S(f(x_i))(1 - S(f(x_i)))| \|\nabla_k f(g_k(x_i))\|_2^2$$
$$\times \mathbb{E}_{\boldsymbol{\xi}_k}[\|\xi_k^{mult} \odot g_k(x_i)\|_2^2 \cos(\nabla_k f(g_k(x_i)), \xi_{mult} \odot g_k(x_i))^2]. \tag{90}$$

Using Theorem 3 and the above results, we obtain:

$$L_n^{NFM} - \frac{1}{n}\sum_{i=1}^n l(f(x_i), y_i)$$

$$\geq \mathbb{E}_k[\epsilon R_1^{(k)} + \epsilon^2 R_2^{(k)} + \epsilon^2 R_2^{add(k)} + \epsilon^2 R_2^{mult(k)} + \epsilon^2 \varphi(\epsilon)] \tag{91}$$

$$\geq \frac{1}{n}\sum_{i=1}^n |S(f(x_i)) - y_i| \|\nabla f(x_i)\|_2 \epsilon_i^{mix} \tag{92}$$

$$+ \frac{1}{2n}\sum_{i=1}^n |S(f(x_i))(1 - S(f(x_i)))| \|\nabla f(x_i)\|_2^2 (\epsilon_i^{mix})^2$$

$$+ \frac{1}{2n}\sum_{i=1}^n |S(f(x_i))(1 - S(f(x_i)))| \|\nabla_k f(g_k(x_i))\|_2^2 \cdot \mathbb{E}_\lambda[(1-\lambda)]^2 \mathbb{E}_{x_r}[\|g_k(x_r)\|_2^2$$

$$\times \cos(\nabla_k f(g_k(x_i)), g_k(x_r))^2] \tag{93}$$

$$+ \frac{1}{2n}\sum_{i=1}^n |S(f(x_i))(1 - S(f(x_i)))| (\epsilon_i^{noise})^2 + \epsilon^2 \varphi(\epsilon), \tag{94}$$

where $\epsilon_i^{mix} := \epsilon \mathbb{E}_{\lambda \sim \tilde{\mathcal{D}}_\lambda}[1 - \lambda] \mathbb{E}_k\left[\frac{\|\nabla_k f(g_k(x_i))\|_2}{\|\nabla f(x_i)\|_2} r_i^{(k)} c_x^{(k)} \sqrt{d_k}\right]$ and

$$(\epsilon_i^{noise})^2 = \epsilon^2 \|\nabla_k f(g_k(x_i))\|_2^2 \left(\sigma_{add}^2 \mathbb{E}_{\boldsymbol{\xi}_k}[\|\xi_k^{add}\|_2^2 \cos(\nabla_k f(g_k(x_i)), \xi_k^{add})^2]\right.$$

$$\left. + \sigma_{mult}^2 \mathbb{E}_{\boldsymbol{\xi}_k}[\|\xi_k^{mult} \odot g_k(x_i)\|_2^2 \cos(\nabla_k f(g_k(x_i)), \xi_k^{mult} \odot g_k(x_i))^2]\right). \tag{95}$$

On the other hand, for any small parameters $\epsilon_i > 0$ and any inputs $z_1, \ldots, z_n$, we can, using a second-order Taylor expansion and then applying our assumptions, compute:

$$\frac{1}{n}\sum_{i=1}^n \max_{\|\delta_i\|_2 \leq \epsilon_i} l(f(z_i + \delta_i), y_i) - \frac{1}{n}\sum_{i=1}^n l(f(z_i), y_i)$$

$$\leq \frac{1}{n}\sum_{i=1}^n |S(f(z_i)) - y_i| \|\nabla f(z_i)\|_2 \epsilon_i + \frac{1}{2n}\sum_{i=1}^n |S(f(z_i))(1 - S(f(z_i)))| \|\nabla f(z_i)\|_2^2 \epsilon_i^2$$

$$+ \frac{1}{n}\sum_{i=1}^n \max_{\|\delta_i\|_2 \leq \epsilon_i} \|\delta_i\|_2^2 \varphi_i'(\delta_i) \tag{96}$$

$$\leq \frac{1}{n}\sum_{i=1}^n |S(f(z_i)) - y_i| \|\nabla f(z_i)\|_2 \epsilon_i + \frac{1}{2n}\sum_{i=1}^n |S(f(z_i))(1 - S(f(z_i)))| \|\nabla f(z_i)\|_2^2 \epsilon_i^2$$

$$+ \frac{1}{n}\sum_{i=1}^n \epsilon_i^2 \varphi_i''(\epsilon_i), \tag{97}$$

where the $\varphi_i'$ are functions such that $\lim_{z \to 0} \varphi_i'(z) = 0$, $\varphi_i''(\epsilon_i) := \max_{\|\delta_i\|_2 \leq \epsilon_i} \varphi_i'(\delta_i)$ and $\lim_{z \to 0} \varphi_i''(z) = 0$.

Combining (94) and (97), we see that

$$L_n^{NFM} \geq \frac{1}{n}\sum_{i=1}^n \max_{\|\delta_i^{mix}\|_2 \leq \epsilon_i^{mix}} l(f(x_i + \delta_i^{mix}), y_i) + L_n^{reg} + \epsilon^2 \varphi(\epsilon) - \frac{1}{n}\sum_{i=1}^n (\epsilon_i^{mix})^2 \varphi_i''(\epsilon_i^{mix}) \tag{98}$$

$$=: \frac{1}{n}\sum_{i=1}^n \max_{\|\delta_i^{mix}\|_2 \leq \epsilon_i^{mix}} l(f(x_i + \delta_i^{mix}), y_i) + L_n^{reg} + \epsilon^2 \phi(\epsilon), \tag{99}$$

where $L_n^{reg}$ is defined in the theorem. Noting that $\lim_{\epsilon \to 0} \phi(\epsilon) = 0$, the proof is done. $\qquad \square$

**Remark 1.** *Had we assumed that $\mathbb{E}_{r \sim \mathcal{D}_x}[g_k(r)] \neq 0$, then the statements of Theorem 4 remain unchanged, but with $(\epsilon_i^{reg})^2$ replaced by*

$$
\begin{aligned}
(\epsilon_i^{reg})^2 = \epsilon^2 \|\nabla_k f(g_k(x_i))\|_2^2 \bigg( & \mathbb{E}_\lambda[(1-\lambda)]^2 \mathbb{E}_{x_r}[\|g_k(x_r)\|_2^2 \cos(\nabla_k f(g_k(x_i)), g_k(x_r))^2] \\
& + \sigma_{add}^2 \mathbb{E}_{\boldsymbol{\xi}}[\|\xi_{add}\|_2^2 \cos(\nabla_k f(g_k(x_i)), \xi_{add})^2] \\
& + \sigma_{mult}^2 \mathbb{E}_{\boldsymbol{\xi}}[\|\xi_{mult} \odot g_k(x_i)\|_2^2 \cos(\nabla_k f(g_k(x_i)), \xi_{mult} \odot g_k(x_i))^2] \bigg) \\
& - \epsilon^2 \mathbb{E}_\lambda[(1-\lambda)]^2 \nabla_k f(g_k(x_i))^T [\mathbb{E}_r g_k(r) g_k(x_i)^T + g_k(x_i) \mathbb{E}_r g_k(r)^T] \nabla_k f(g_k(x_i)).
\end{aligned}
\tag{100}
$$

## C  NFM THROUGH THE LENS OF IMPLICIT REGULARIZATION AND CLASSIFICATION MARGIN

First, we define classification margin at the input level. We shall show that minimizing the NFM loss can lead to an increase in the classification margin, and therefore improve model robustness in this sense.

**Definition 2** (Classification Margin). *The classification margin of a training input-label sample $s_i := (x_i, c_i)$ measured by the Euclidean metric $d$ is defined as the radius of the largest $d$-metric ball in $\mathcal{X}$ centered at $x_i$ that is contained in the decision region associated with the class label $c_i$, i.e., it is: $\gamma^d(s_i) = \sup\{a : d(x_i, x) \leq a \Rightarrow g(x) = c_i \ \forall x\}$.*

Intuitively, a larger classification margin allows a classifier to associate a larger region centered on a point $x_i$ in the input space to the same class. This makes the classifier less sensitive to input perturbations, and a perturbation of $x_i$ is still likely to fall within this region, keeping the classifier prediction. In this sense, the classifier becomes more robust. In the typical case, the networks are trained by a loss (cross-entropy) that promotes separation of different classes in the network output. This, in turn, maximizes a certain notion of score of each training sample (Sokolić et al., 2017).

**Definition 3** (Score). *For an input-label training sample $s_i = (x_i, c_i)$, we define its score as $o(s_i) = \min_{j \neq c_i} \sqrt{2}(e_{c_i} - e_j)^T f(x_i) \geq 0$, where $e_i \in \mathbb{R}^K$ is the Kronecker delta vector (one-hot vector) with $e_i^i = 1$ and $e_i^j = 0$ for $i \neq j$.*

A positive score implies that at the network output, classes are separated by a margin that corresponds to the score. A large score may not imply a large classification margin, but score can be related to classification margin via the following bound.

**Proposition 1.** *Assume that the score $o(s_i) > 0$ and let $k \in \mathcal{S}$. Then, the classification margin for the training sample $s_i$ can be lower bounded as:*

$$
\gamma^d(s_i) \geq \frac{C(s_i)}{\sup_{x \in conv(\mathcal{X})} \|\nabla_k f(g_k(x))\|_2},
\tag{101}
$$

*where $C(s_i) = o(s_i) / \sup_{x \in conv(\mathcal{X})} \|\nabla g_k(x)\|_2$.*

Since NFM implicitly reduces the feature-output Jacobians $\nabla_k f$ (including the input-output Jacobian) according to the mixup level and noise levels (see Proposition 3), this, together with Theorem 1, suggests that applying NFM implicitly increases the classification margin, thereby making the model more robust to input perturbations. We note that a similar, albeit more involved, bound can also be obtained for the all-layer margin, a more refined version of classification margin introduced in (Wei & Ma, 2019b), and the conclusion that applying NFM implicitly increases the margin also holds.

We now prove the proposition.

*Proof of Proposition 1.* Note that, for any $k \in \mathcal{S}$, $\nabla f(x) = \nabla_k f(g_k(x)) \nabla g_k(x)$ by the chain rule, and so

$$
\|\nabla f(x)\|_2 \leq \|\nabla_k f(g_k(x))\|_2 \|\nabla g_k(x)\|_2
\tag{102}
$$

$$
\leq \left( \sup_{x \in conv(\mathcal{X})} \|\nabla_k f(g_k(x))\|_2 \right) \left( \sup_{x \in conv(\mathcal{X})} \|\nabla g_k(x)\|_2 \right).
\tag{103}
$$

The statement in the proposition follows from a straightforward application of Theorem 4 in (Sokolić et al., 2017) together with the above bound. □

## D  NFM THROUGH THE LENS OF PROBABILISTIC ROBUSTNESS

Since the main novelty of NFM lies in the introduction of noise injection, it would be insightful to isolate the robustness boosting benefits of injecting noise on top of manifold mixup. We shall demonstrate the isolated benefit in this section.

The key idea is based on the observation that manifold mixup produces minibatch outputs that lie in the convex hull of the feature space at each iteration. Therefore, for $k \in \mathcal{S}$, $NFM(k)$ can be viewed as injecting noise to the layer $k$ features sampled from some distribution over $conv(g_k(\mathcal{X}))$, and so the $NFM(k)$ neural network $F_k$ can be viewed as a probabilistic mapping from $conv(g_k(\mathcal{X}))$ to $\mathcal{P}(\mathcal{Y})$, the space of probability distributions on $\mathcal{Y}$.

To isolate the benefit of noise injection, we adapt the approach of (Pinot et al., 2019a; 2021) to our setting to show that the Gaussian noise injection procedure in NFM robustifies manifold mixup in a probabilistic sense. At its core, this probabilistic notion of robustness amounts to making the model locally Lipschitz with respect to some distance on the input and output space, ensuring that a small perturbation in the input will not lead to large changes (as measured by some probability metric) in the output. Interestingly, it is related to a notion of differential privacy (Lecuyer et al., 2019; Dwork et al., 2014), as formalized in (Pinot et al., 2019b).

We now formalize this probabilistic notion of robustness.

Let $p > 0$. We say that a standard model $f : \mathcal{X} \to \mathcal{Y}$ is $\alpha_p$-robust if for any $(x, y) \sim \mathcal{D}$ such that $f(x) = y$, one has, for any data perturbation $\tau \in \mathcal{X}$,

$$\|\tau\|_p \leq \alpha_p \implies f(x) = f(x + \tau). \tag{104}$$

Analogous definition can be formulated when output of the model is distribution-valued.

**Definition 4** (Probabilistic robustness). *A probabilistic model $F : \mathcal{X} \to \mathcal{P}(\mathcal{Y})$ is called $(\alpha_p, \epsilon)$-robust with respect to D if, for any $x, \tau \in \mathcal{X}$, one has*

$$\|\tau\|_p \leq \alpha_p \implies D(F(x), F(x + \tau)) \leq \epsilon, \tag{105}$$

*where D is a metric or divergence between two probability distributions.*

We refer to the probabilistic model (built on top of a manifold mixup classifier) that injects Gaussian noise to the layer $k$ features as *probabilistic FM model*, and we denote it by $F^{noisy(k)} : conv(g_k(\mathcal{X})) \to \mathcal{P}(\mathcal{Y})$. We denote $G$ as the classifier constructed from $F^{noisy(k)}$, i.e., $G : x \mapsto \arg\max_{j \in [K]} [F^{noisy(k)}]^j(x)$.

In the sequel, we take $D$ to be the total variation distance $D_{TV}$, defined as:

$$D_{TV}(P, Q) := \sup_{S \subset \mathcal{X}} |P(S) - Q(S)|, \tag{106}$$

for any two distributions $P$ and $Q$ over $\mathcal{X}$. Recall that if $P$ and $Q$ have densities $\rho_p$ and $\rho_q$ respectively, then the total variation distance is half of the $L^1$ distance, i.e., $D_{TV}(P, Q) = \frac{1}{2} \int_{\mathcal{X}} |\rho_p(x) - \rho_q(x)| dx$. The choice of the distance depends on the problem on hand and will give rise to different notions of robustness. One could also consider other statistical distances such as the Wasserstein distance and Renyi divergence, which can be related to total variation (see (Pinot et al., 2021; Gibbs & Su, 2002) for details).

Before presenting our main result in this section, we need the following notation. Let $\Sigma(x) := \sigma^2_{add} I + \sigma^2_{mult} xx^T$. For $x, \tau \in \mathcal{X}$, let $\Pi_x$ be a $d_k$ by $d_k - 1$ matrix whose columns form a basis for the subspace orthogonal to $g_k(x + \tau) - g_k(x)$, and $\{\rho_i(g_k(x), \tau)\}_{i \in [d_k - 1]}$ be the eigenvalues of $(\Pi_x^T \Sigma(g_k(x))\Pi_x)^{-1}\Pi_x^T \Sigma(g_k(x + \tau))\Pi_x - I$. Also, let $[F]^{topk}(x)$ denote the $k$th highest value of the entries in the vector $F(x)$.

Viewing an $NFM(k)$ classifier as a probabilistic FM classifier, we have the following result.

**Theorem 5** (Gaussian noise injection robustifies FM classifiers). *Let $k \in \mathcal{S}$, $d_k > 1$, and assume that $g_k(x)g_k(x)^T \geq \beta_k^2 I > 0$ for all $x \in conv(\mathcal{X})$ for some constant $\beta_k$. Then, $F^{noisy(k)}$ is $(\alpha_p, \epsilon_k(p, d, \alpha_p, \sigma_{add}, \sigma_{mult}))$-robust with respect to $D_{TV}$ against $l_p$ adversaries, with*

$$\epsilon_k(p, d, \alpha_p, \sigma_{add}, \sigma_{mult}) = \frac{9}{2} \min\{1, \max\{A, B\}\}, \tag{107}$$

*where*

$$A = A_p(\alpha_p) \frac{\sigma_{mult}^2}{\sigma_{add}^2 + \sigma_{mult}^2 \beta_k^2} \left( \left\| \int_0^1 \nabla g_k(x + t\tau)dt \right\|_2^2 + 2\|g_k(x)\|_2 \left\| \int_0^1 \nabla g_k(x + t\tau)dt \right\|_2 \right), \tag{108}$$

$$B = B_k(\tau) \frac{\alpha_p(\mathbb{1}_{p \in (0,2]} + d^{1/2 - 1/p} \mathbb{1}_{p \in (2,\infty)} + \sqrt{d} \mathbb{1}_{p=\infty})}{\sqrt{\sigma_{add}^2 + \sigma_{mult}^2 \beta_k^2}}, \tag{109}$$

*with*

$$A_p(\alpha_p) = \begin{cases} \alpha_p \mathbb{1}_{\alpha_p < 1} + \alpha_p^2 \mathbb{1}_{\alpha_p \geq 1}, & \text{if } p \in (0, 2], \\ d^{1/2 - 1/p}(\alpha_p \mathbb{1}_{\alpha_p < 1} + \alpha_p^2 \mathbb{1}_{\alpha_p \geq 1}), & \text{if } p \in (2, \infty), \\ \sqrt{d}(\alpha_p \mathbb{1}_{\alpha_p < 1} + \alpha_p^2 \mathbb{1}_{\alpha_p \geq 1}), & \text{if } p = \infty, \end{cases} \tag{110}$$

*and*

$$B_k(\tau) = \sup_{x \in conv(\mathcal{X})} \left( \left\| \int_0^1 \nabla g_k(x + t\tau)dt \right\|_2 \cdot \sqrt{\sum_{i=1}^{d_k - 1} \rho_i^2(g_k(x), \tau)} \right). \tag{111}$$

*Moreover, if $x \in \mathcal{X}$ is such that $[F^{noisy(k)}]^{top1}(x) \geq [F^{noisy(k)}]^{top2}(x) + 2\epsilon(p, d, \alpha_p, \sigma_{add}, \sigma_{mult})$, then for any $\tau \in \mathcal{X}$, we have*

$$\|\tau\|_p \leq \alpha \implies G(x) = G(x + \tau), \tag{112}$$

*for any $p > 0$.*

Theorem 5 implies that we can inject Gaussian noise into the feature mixup representation to improve robustness of FM classifiers in the sense of Definition 4, while keeping track of maximal loss in accuracy incurred under attack, by tuning the noise levels $\sigma_{add}$ and $\sigma_{mult}$. To illustrate this, suppose that $\sigma_{mult} = 0$ and consider the case of $p = 2$, in which case $A = 0$, $B \sim \alpha_2/\sigma_{add}$ and so injecting additive Gaussian noise can help controlling the change in the model output, keeping the classifier's prediction, when the data perturbation is of size $\alpha_2$.

We now prove Theorem 5. Before this, we need the following lemma.

**Lemma 3.** *Let $x_1 := z \in \mathbb{R}^{d_k}$ and $x_2 := z + \tau \in \mathbb{R}^{d_k}$, with $\tau > 0$ and $d_k > 1$, and $\Sigma(x) := \sigma_{add}^2 I + \sigma_{mult}^2 x x^T \geq (\sigma_{add}^2 + \sigma_{mult}^2 \beta^2)I > 0$, for some constant $\beta$, for all $x$. Let $\Pi$ be a $d_k$ by $d_k - 1$ matrix whose columns form a basis for the subspace orthogonal to $\tau$, and let $\rho_1(z, \tau), \ldots, \rho_{d_k - 1}(z, \tau)$ denote the eigenvalues of $(\Pi^T \Sigma(x_1)\Pi)^{-1}\Pi^T \Sigma(x_2)\Pi - I$.*

*Define the function $C(x_1, x_2, \Sigma) := \max\{A, B\}$, where*

$$A = \frac{\sigma_{mult}^2}{\sigma_{add}^2 + \sigma_{mult}^2 \beta^2} (\|\tau\|_2^2 + 2\tau^T z), \tag{113}$$

$$B = \frac{\|\tau\|_2}{\sqrt{\sigma_{add}^2 + \sigma_{mult}^2 \beta^2}} \sqrt{\sum_{i=1}^{d_k - 1} \rho_i^2(z, \tau)}. \tag{114}$$

*Then, the total variation distance between $\mathcal{N}(x_1, \Sigma(x_1))$ and $\mathcal{N}(x_2, \Sigma(x_2))$ admits the following bounds:*

$$\frac{1}{200} \leq \frac{D_{TV}(\mathcal{N}(x_1, \Sigma(x_1)), \mathcal{N}(x_2, \Sigma(x_2)))}{\min\{1, C(x_1, x_2, \Sigma)\}} \leq \frac{9}{2}. \tag{115}$$

*Proof of Lemma 3.* The result follows from a straightforward application of Theorem 1.2 in (Devroye et al., 2018), which provides bounds on the total variation distance between Gaussians with different means and covariances. $\square$

With this lemma in hand, we now prove Theorem 5.

*Proof of Theorem 5.* We denote the noise injection procedure by the map $\mathcal{I} : x \to \mathcal{N}(x, \Sigma(x))$, where $\Sigma(x) = \sigma_{add}^2 I + \sigma_{mult}^2 x x^T$.

Let $x \in \mathcal{X}$ be a test datapoint and $\tau \in \mathcal{X}$ be a data perturbation such that $\|\tau\|_p \leq \alpha_p$ for $p > 0$.

Note that

$$D_{TV}(F_k(\mathcal{I}(g_k(x))), F_k(\mathcal{I}(g_k(x+\tau)))) \leq D_{TV}(\mathcal{I}(g_k(x)), \mathcal{I}(g_k(x+\tau))) \tag{116}$$

$$\leq D_{TV}(\mathcal{I}(g_k(x)), \mathcal{I}(g_k(x) + g_k(x+\tau) - g_k(x))) \tag{117}$$

$$= D_{TV}(\mathcal{I}(g_k(x)), \mathcal{I}(g_k(x) + \tau_k)) \tag{118}$$

$$\leq \frac{9}{2} \min\{1, \Phi(g_k(x), \tau_k, \sigma_{add}, \sigma_{mult}, \beta_k)\}, \tag{119}$$

where $\tau_k := g_k(x+\tau) - g_k(x) = \left( \int_0^1 \nabla g_k(x+t\tau) dt \right) \tau$ by the generalized fundamental theorem of calculus, and

$$\Phi(g_k(x), \tau_k, \sigma_{add}, \sigma_{mult}, \beta_k)$$

$$:= \max \left\{ \frac{\sigma_{mult}^2}{\sigma_{add}^2 + \sigma_{mult}^2 \beta_k^2} (\|\tau_k\|_2^2 + 2\langle \tau_k, g_k(x) \rangle), \frac{\|\tau_k\|_2}{\sqrt{\sigma_{add}^2 + \sigma_{mult}^2 \beta_k^2}} \sqrt{\sum_{i=1}^{d_k-1} \rho_i^2(g_k(x), \tau)} \right\}, \tag{120}$$

where the $\rho_i(g_k(x), \tau)$ are the eigenvalues given in the theorem.

In the first line above, we have used the data preprocessing inequality (Theorem 6 in (Pinot et al., 2021)), and the last line follows from applying Lemma 3 together with the assumption that $g_k(x) g_k(x)^T \geq \beta_k^2 > 0$ for all $x$.

Using the bounds

$$\|\tau_k\|_2 \leq \left\| \int_0^1 \nabla g_k(x+t\tau) dt \right\|_2 \|\tau\|_2 \tag{121}$$

and

$$|\langle \tau_k, g_k(x) \rangle| \leq \|g_k(x)\|_2 \left\| \int_0^1 \nabla g_k(x+t\tau) dt \right\|_2 \|\tau\|_2, \tag{122}$$

we have

$$\Phi(g_k(x), \tau_k, \sigma_{add}, \sigma_{mult}, \beta_k) \leq \max\{A, B\}, \tag{123}$$

where

$$A = \frac{\sigma_{mult}^2}{\sigma_{add}^2 + \sigma_{mult}^2 \beta_k^2} \left( \left\| \int_0^1 \nabla g_k(x+t\tau) dt \right\|_2^2 \|\tau\|_2^2 + 2\|g_k(x)\|_2 \left\| \int_0^1 \nabla g_k(x+t\tau) dt \right\|_2 \|\tau\|_2 \right) \tag{124}$$

and

$$B = \frac{\left\| \int_0^1 \nabla g_k(x+t\tau) dt \right\|_2 \|\tau\|_2}{\sqrt{\sigma_{add}^2 + \sigma_{mult}^2 \beta_k^2}} \sqrt{\sum_{i=1}^{d_k-1} \rho_i^2(g_k(x), \tau)} \tag{125}$$

$$\leq \sup_{x \in conv(\mathcal{X})} \left( \left\| \int_0^1 \nabla g_k(x+t\tau) dt \right\|_2 \cdot \sqrt{\sum_{i=1}^{d_k-1} \rho_i^2(g_k(x), \tau)} \right) \frac{\|\tau\|_2}{\sqrt{\sigma_{add}^2 + \sigma_{mult}^2 \beta_k^2}} \tag{126}$$

$$=: B_k(\tau) \frac{\|\tau\|_2}{\sqrt{\sigma_{add}^2 + \sigma_{mult}^2 \beta_k^2}}. \tag{127}$$

The first statement of the theorem then follows from the facts that $\|\tau\|_2 \le \|\tau\|_p \le \alpha_p$ for $p \in (0, 2]$, $\|\tau\|_2 \le d^{1/2-1/q}\|\tau\|_q \le d^{1/2-1/q}\alpha_q$ for $q > 2$, and $\|\tau\|_2 \le \sqrt{d}\|\tau\|_\infty \le \sqrt{d}\alpha_\infty$ for any $\tau \in \mathbb{R}^d$. In particular, these imply that $A \le CA_p$, where

$$A_p = \begin{cases} \alpha_p \mathbb{1}_{\alpha_p<1} + \alpha_p^2 \mathbb{1}_{\alpha_p \ge 1}, & \text{if } p \in (0, 2], \\ d^{1/2-1/p}(\alpha_p \mathbb{1}_{\alpha_p<1} + \alpha_p^2 \mathbb{1}_{\alpha_p \ge 1}), & \text{if } p \in (2, \infty), \\ \sqrt{d}(\alpha_p \mathbb{1}_{\alpha_p<1} + \alpha_p^2 \mathbb{1}_{\alpha_p \ge 1}), & \text{if } p = \infty, \end{cases} \tag{128}$$

and

$$C := \frac{\sigma_{mult}^2}{\sigma_{add}^2 + \sigma_{mult}^2 \beta_k^2} \left( \left\| \int_0^1 \nabla g_k(x + t\tau) dt \right\|_2^2 + 2\|g_k(x)\|_2 \left\| \int_0^1 \nabla g_k(x + t\tau) dt \right\|_2 \right). \tag{129}$$

The last statement in the theorem essentially follows from Proposition 3 in (Pinot et al., 2021).  $\square$

## E  ON GENERALIZATION BOUNDS FOR NFM

Let $\mathcal{F}$ be the family of mappings $x \mapsto f(x)$ and $Z_n := ((x_i, y_i))_{i \in [n]}$. Given a loss function $l$, the Rademacher complexity of the set $l \circ \mathcal{F} := \{(x, y) \mapsto l(f(x), y) : f \in \mathcal{F}\}$ is defined as:

$$R_n(l \circ \mathcal{F}) := \mathbb{E}_{Z_n, \sigma} \left[ \sup_{f \in \mathcal{F}} \frac{1}{n} \sum_{i=1}^n \sigma_i l(f(x_i), y_i) \right], \tag{130}$$

where $\sigma := (\sigma_1, \ldots, \sigma_n)$, with the $\sigma_i$ independent uniform random variables taking values in $\{-1, 1\}$.

Following (Lamb et al., 2019), we can derive the following generalization bound for the NFM loss function, i.e., the upper bound on the difference between the expected error on unseen data and the NFM loss. This bound shows that NFM can reduce overfitting and give rise to improved generalization.

**Theorem 6** (Generalization bound for the NFM loss). *Assume that the loss function $l$ satisfies $|l(x, y) - l(x', y)| \le M$ for all $x, x'$ and $y$. Then, for every $\delta > 0$, with probability at least $1 - \delta$ over a draw of $n$ i.i.d. samples $\{(x_i, y_i)\}_{i=1}^n$, we have the following generalization bound: for all maps $f \in \mathcal{F}$,*

$$\mathbb{E}_{x,y}[l(f(x), y)] - L_n^{NFM} \le 2R_n(l \circ \mathcal{F}) + 2M\sqrt{\frac{\ln(1/\delta)}{2n}} - Q_\epsilon(f), \tag{131}$$

*where*

$$Q_\epsilon(f) = \mathbb{E}[\epsilon R_1^{(k)} + \epsilon^2 \tilde{R}_2^{(k)} + \epsilon^2 \tilde{R}_3^{(k)}] + \epsilon^2 \varphi(\epsilon), \tag{132}$$

*for some function $\varphi$ such that $\lim_{x \to \infty} \varphi(x) = 0$.*

To compare the generalization behavior of NFM with that without using NFM, we also need the following generalization bound for the standard loss function.

**Theorem 7** (Generalization bound for the standard loss). *Assume that the loss function $l$ satisfies $|l(x, y) - l(x', y)| \le M$ for all $x, x'$ and $y$. Then, for every $\delta > 0$, with probability at least $1 - \delta$ over a draw of $n$ i.i.d. samples $\{(x_i, y_i)\}_{i=1}^n$, we have the following generalization bound: for all maps $f \in \mathcal{F}$,*

$$\mathbb{E}_{x,y}[l(f(x), y)] - L_n^{std} \le 2R_n(l \circ \mathcal{F}) + 2M\sqrt{\frac{\ln(1/\delta)}{2n}}. \tag{133}$$

By comparing the above two theorems and following the argument of (Lamb et al., 2019), we see that the generalization benefit of NFM comes from two mechanisms. The first mechanism is based on the term $Q_\epsilon(f)$. Assuming that the Rademacher complexity term is the same for both methods, then NFM has a better generalization bound than that of standard method if $Q_\epsilon(f) > 0$. The second mechanism is based on the Rademacher complexity term $R_n(l \circ \mathcal{F})$. For certain families of neural networks, this term can be bounded by the norms of the hidden layers of the network and the norms of the Jacobians of each layer with respect to all previous layers (Wei & Ma, 2019a;b). Therefore,

this term differs for the case of training using NFM and the case of standard training. Since NFM implicitly reduces the feature-output Jacobians (see Theorem 3), we can argue that NFM leads to a smaller Rademacher complexity term and hence a better generalization bound.

We now prove Theorem 6. The proof of Theorem 7 follows the same argument as that of Theorem 6.

*Proof of Theorem 6.* Let $Z_n := \{(x_i, y_i)\}_{i \in [n]}$ and $Z'_n := \{(x'_i, y'_i)\}_{i \in [n]}$ be two test datasets, where $Z'_n$ differs from $Z_n$ by exactly one point of an arbitrary index $i_0$.

Denote $GE(Z_n) := \sup_{f \in \mathcal{F}} \mathbb{E}_{x,y}[l(f(x), y)] - L_n^{NFM}$, where $L_n^{NFM}$ is computed using the dataset $Z_n$, and likewise for $GE(Z'_n)$. Then,

$$GE(Z'_n) - GE(Z_n) \leq \frac{M(2n-1)}{n^2} \leq \frac{2M}{n}, \tag{134}$$

where we have used the fact that $L_n^{NFM}$ has $n^2$ terms and there are $2n-1$ different terms for $Z_n$ and $Z'_n$. Similarly, we have $GE(Z_n) - GE(Z'_n) \leq \frac{2M}{n}$.

Therefore, by McDiarmid's inequality, for any $\delta > 0$, with probability at least $1 - \delta$,

$$GE(Z_n) \leq \mathbb{E}_{Z_n}[GE(Z_n)] + 2M\sqrt{\frac{\ln(1/\delta)}{2n}}. \tag{135}$$

Applying Theorem 3, we have

$$GE(Z_n) \leq \mathbb{E}_{Z_n}\left[\sup_{f \in \mathcal{F}} \mathbb{E}_{Z'_n}\left[\frac{1}{n}\sum_{i=1}^{n} l(f(x'_i), y'_i)\right] - L_n^{NFM}\right] + 2M\sqrt{\frac{\ln(1/\delta)}{2n}} \tag{136}$$

$$= \mathbb{E}_{Z_n}\left[\sup_{f \in \mathcal{F}} \mathbb{E}_{Z'_n}\left[\frac{1}{n}\sum_{i=1}^{n} l(f(x'_i), y'_i)\right] - \frac{1}{n}\sum_{i=1}^{n} l(f(x_i), y_i)\right] - Q_\epsilon(f)$$

$$+ 2M\sqrt{\frac{\ln(1/\delta)}{2n}} \tag{137}$$

$$\leq \mathbb{E}_{Z_n, Z'_n}\left[\sup_{f \in \mathcal{F}} \frac{1}{n}\sum_{i=1}^{n}(l(f(x'_i), y'_i) - l(f(x_i), y_i))\right] - Q_\epsilon(f) + 2M\sqrt{\frac{\ln(1/\delta)}{2n}} \tag{138}$$

$$\leq \mathbb{E}_{Z_n, Z'_n, \sigma}\left[\sup_{f \in \mathcal{F}} \frac{1}{n}\sum_{i=1}^{n} \sigma_i(l(f(x'_i), y'_i) - l(f(x_i), y_i))\right] - Q_\epsilon(f) + 2M\sqrt{\frac{\ln(1/\delta)}{2n}} \tag{139}$$

$$\leq 2\mathbb{E}_{Z_n, \sigma}\left[\sup_{f \in \mathcal{F}} \frac{1}{n}\sum_{i=1}^{n} \sigma_i l(f(x_i), y_i)\right] - Q_\epsilon(f) + 2M\sqrt{\frac{\ln(1/\delta)}{2n}} \tag{140}$$

$$= 2R_n(l \circ \mathcal{F}) - Q_\epsilon(f) + 2M\sqrt{\frac{\ln(1/\delta)}{2n}}, \tag{141}$$

where (136) uses the definition of $GE(Z_n)$, (137) uses $\pm \frac{1}{n}\sum_{i=1}^{n} l(f(x_i), y_i)$ inside the expectation and the linearity of expectation, (138) follows from the Jensen's inequality and the convexity of the supremum, (139) follows from the fact that $\sigma_i(l(f(x'_i), y'_i) - l(f(x_i), y_i))$ and $l(f(x'_i), y'_i) - l(f(x_i), y_i)$ have the same distribution for each $\sigma_i \in \{-1, 1\}$ (since $Z_n, Z'_n$ are drawn i.i.d. with the same distribution), and (140) follows from the subadditivity of supremum.

The bound in the theorem then follows from the above bound. $\square$

## F  ADDITIONAL EXPERIMENTS AND DETAILS

### F.1  INPUT PERTURBATIONS

We consider the following three types of data perturbations during inference time:

- *White noise perturbations* are constructed as $\tilde{x} = x + \Delta x$, where the additive noise is sampled from a Gaussian distribution $\Delta x \sim \mathcal{N}(0, \sigma)$. This perturbation strategy emulates measurement errors that can result from data acquisition with poor sensors (where $\sigma$ corresponds to the severity of these errors).

- *Salt and pepper perturbations* emulate defective pixels that result from converting analog signals to digital signals. The noise model takes the form $\mathbb{P}(\tilde{X} = X) = 1 - \gamma$, and $\mathbb{P}(\tilde{X} = \max) = \mathbb{P}(\tilde{X} = \min) = \gamma/2$, where $\tilde{X}(i, j)$ denotes the corrupted image and $\min, \max$ denote the minimum and maximum pixel values, respectively. $\gamma$ parameterizes the proportion of defective pixels.

- *Adversarial perturbations* are "worst-case" non-random perturbations that maximize the loss $\ell(g^\delta(X + \Delta X), y)$ subject to the constraint $\|\Delta X\| \leq r$ on the norm of the perturbation. We consider the projected gradient decent for constructing these perturbations (Madry et al., 2017).

### F.2  ILLUSTRATION OF THE EFFECTS OF NFM ON TOY DATASETS

We consider a binary classification task for the noise corrupted 2D dataset whose data points form two concentric circles. Points on the same circle corresponds to the same label class. We generate 500 samples, setting the scale factor between inner and outer circle to be 0.05 and adding Gaussian noise with zero mean and standard deviation of 0.3 to the samples. Fig. 8 shows the training and test data points. We train a fully connected feedforward neural network that has four layers with the ReLU activation functions on these data, using 300 points for training and 200 for testing. All models are trained with Adam and learning rate 0.1, and the seed is fixed across all experiments. Note that the learning rate can be considered as a temperature parameter which introduces some amount of regularization itself. Hence, we choose a learning rate that is large for this problem to better illustrate the regularization effects imposed by the different schemes that we consider.

Fig. 2 illustrates how different regularization strategies affect the decision boundaries of the neural network classifier. The decision boundaries and the test accuracy indicate that white noise injections and dropout (we explore dropout rates in the range $[0.0, 0.9]$ and we finds that $0.2$ yields the best performance) introduce a favorable amount of regularization. Most notably is the effect of weight decay (we use $9e-3$), i.e., the decision boundary is nicely smoothed and the test accuracy is improved. In contrast, the simple mixup data augmentation scheme shows no benefits here, whereas manifold mixup is improving the predictive accuracy considerably. Combining mixup (manifold mixup) with noise injections yields the best performance in terms of both smoothness of the decision boundary and predictive accuracy. Indeed, NFM is outperforming all other methods here.

The performance could be further improved by combining NFM with weight decay or dropout. This shows that there are interaction effects between different regularization schemes. In practice, when

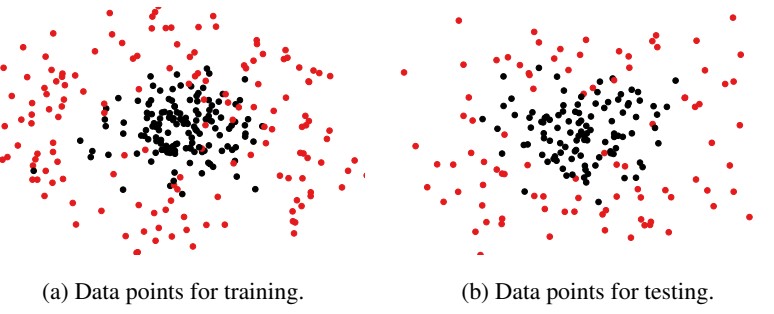

(a) Data points for training.     (b) Data points for testing.

Figure 8: The toy dataset in $\mathbb{R}^2$ that we use for binary classification.

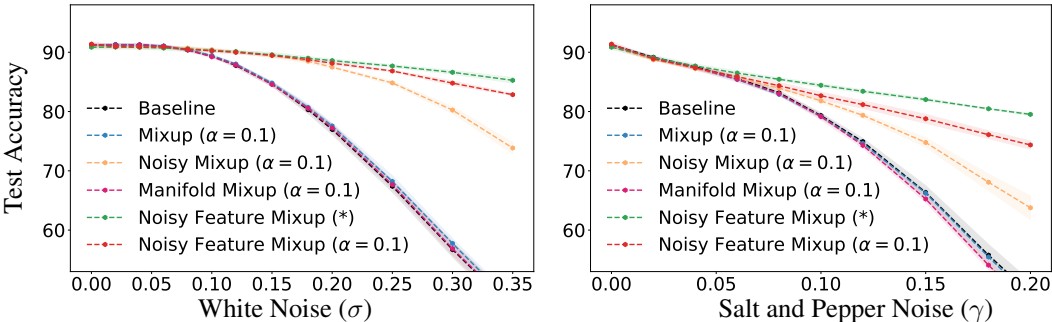

Figure 9: Vision transformers evaluated on CIFAR-10 with different training schemes.

Table 4: Robustness of Wide-ResNet-18 w.r.t. white noise ($\sigma$) and salt and pepper ($\gamma$) perturbations evaluated on CIFAR-100. The results are averaged over 5 models trained with different seed values.

| Scheme | Clean (%) | $\sigma$ (%) | | | $\gamma$ (%) | | |
|---|---|---|---|---|---|---|---|
| | | 0.1 | 0.2 | 0.3 | 0.08 | 0.12 | 0.2 |
| Baseline | 91.3 | 89.4 | 77.0 | 56.7 | 83.2 | 74.6 | 48.6 |
| Mixup ($\alpha = 0.1$) Zhang et al. (2017) | 91.2 | 89.5 | 77.6 | 57.7 | 82.9 | 74.6 | 48.6 |
| Mixup ($\alpha = 0.2$) Zhang et al. (2017) | 91.2 | 89.2 | 77.8 | 58.9 | 82.6 | 74.5 | 47.9 |
| Noisy Mixup ($\alpha = 0.1$) Yang et al. (2020b) | 90.9 | 90.4 | 87.5 | 80.2 | 84.0 | 79.4 | 63.8 |
| Noisy Mixup ($\alpha = 0.2$) Yang et al. (2020b) | 90.9 | 90.4 | 87.4 | 79.8 | 83.8 | 79.3 | 63.4 |
| Manifold Mixup ($\alpha = 0.1$) Verma et al. (2019) | 91.2 | 89.2 | 77.2 | 56.9 | 83.0 | 74.3 | 47.1 |
| Manifold Mixup ($\alpha = 1.0$) Verma et al. (2019) | 90.2 | 88.4 | 76.0 | 55.1 | 81.3 | 71.4 | 42.7 |
| Manifold Mixup ($\alpha = 2.0$) Verma et al. (2019) | 89.0 | 87.0 | 74.3 | 53.7 | 79.8 | 70.3 | 41.9 |
| Noisy Feature Mixup ($\alpha = 0.1$) | **91.4** | **90.2** | **88.2** | **84.8** | **84.4** | **81.2** | **74.4** |
| Noisy Feature Mixup ($\alpha = 1.0$) | 89.8 | 89.1 | 86.6 | 82.7 | 82.5 | 79.0 | 71.4 |
| Noisy Feature Mixup ($\alpha = 2.0$) | 88.4 | 87.6 | 84.6 | 80.1 | 80.4 | 76.5 | 68.6 |

one trains deep neural networks, different regularization strategies are considered as knobs that are fine-tuned. From this perspective, NFM provides additional knobs to further improve a model.

### F.3 ADDITIONAL RESULTS FOR VISION TRANSFORMERS

Here we consider compact vision transformer (ViT-lite) with 7 attention layers and 4 heads (Hassani et al., 2021). Fig. 9 (left) compares vision transformers trained with different data augmentation strategies. Again, NFM improves the robustness of the models while achieving state-of-the-art accuracy when evaluated on clean data. However, mixup and manifold mixup do not boost the robustness. Further, Fig. 9 (right) shows that that the vision transformer is less sensitive to salt and pepper perturbations as compared to the ResNet model. These results are consistent with the high robustness properties of transformers recently reported in Shao et al. (2021); Paul & Chen (2021). Table 4 provides additional results for different $\alpha$ values.

Table 4 shows results for vision transformers trained with different data augmentation schemes and different values of $\alpha$. It can be seen that NFM with $\alpha = 0.1$ helps to improve the predictive accuracy on clean data while also improving the robustness of the models. For example, the model trained with NFM shows about a $25\%$ improvement compared to the baseline model when faced with salt and paper perturbations ($\gamma = 0.2$). Further, our results indicate that larger values of $\alpha$ have a negative effect on the generalization performance of vision transformer.

### F.4 ABLATION STUDY

In Table 5 we provide a detailed ablation study where we vary several knobs. First, we can see that just injecting noise helps to improve robustness, but the test accuracy is only marginally improving. On the other hand, just mixing inputs and hidden features improves the testing performance of the model, but it does not significantly improve the robustness of a model. In contrast, the NFM scheme combines best of both worlds and shows that both accuracy and robustness can be increased. Varying the noise levels indicate that there is a trade-off between test accuracy on clean data and robustness to

perturbations. We also vary the mixup parameter $\alpha$ to show that the good performance is consistent across a range of different values.

Table 5: Ablation study using Wide-ResNet-18 trained and evaluated on CIFAR-100.

| Mixup | Manifold | Noise Injections | $\alpha$ | Noise Levels | | Clean (%) | $\sigma$ (%) | | | $\gamma$ (%) | | |
|---|---|---|---|---|---|---|---|---|---|---|---|---|
| | | | | $\sigma_{add}$ | $\sigma_{mult}$ | | 0.1 | 0.25 | 0.5 | 0.06 | 0.1 | 0.15 |
| ✗ | ✗ | ✗ | - | 0 | 0 | 76.9 | 64.6 | 42.0 | 23.5 | 58.1 | 39.8 | 15.1 |
| ✗ | ✗ | ✓ | - | 0.4 | 0.2 | 78.1 | 76.2 | 65.7 | 46.6 | 70.0 | 58.8 | 28.4 |
| ✓ | ✗ | ✗ | 1 | 0 | 0 | 80.3 | 72.5 | 54.0 | 33.4 | 62.5 | 43.8 | 16.2 |
| ✓ | ✗ | ✓ | 1 | 0.4 | 0.2 | 78.9 | 78.6 | 66.6 | 46.7 | 66.6 | 53.4 | 25.9 |
| ✓ | ✓ | ✗ | 0.2 | 0 | 0 | 79.7 | 70.6 | 46.6 | 25.3 | 62.1 | 43.0 | 15.2 |
| ✓ | ✓ | ✗ | 1 | 0 | 0 | 79.7 | 70.5 | 45.0 | 23.8 | 62.1 | 42.8 | 14.8 |
| ✓ | ✓ | ✗ | 2 | 0 | 0 | 79.2 | 69.3 | 43.8 | 23.0 | 62.8 | 44.2 | 16.0 |
| ✓ | ✓ | ✓ | 1 | 0.1 | 0.1 | **81.0** | 76.2 | 56.6 | 36.4 | 66.8 | 49.7 | 21.4 |
| ✓ | ✓ | ✓ | 0.2 | 0.4 | 0.2 | 80.6 | 79.2 | 70.2 | 51.7 | 71.5 | 60.4 | 30.3 |
| ✓ | ✓ | ✓ | 1 | 0.4 | 0.2 | 80.9 | **80.1** | 72.1 | 55.3 | 72.8 | 62.1 | 34.4 |
| ✓ | ✓ | ✓ | 2 | 0.4 | 0.2 | 80.7 | 80.0 | 71.5 | 53.9 | 72.7 | 62.7 | 36.6 |
| ✓ | ✓ | ✓ | 1 | 0.8 | 0.4 | 80.3 | **80.1** | **75.5** | **66.4** | **74.3** | **66.5** | **44.6** |

### F.5 ADDITIONAL RESULTS FOR RESNETS WITH HIGHER LEVELS OF NOISE INJECTIONS

In the experiments in Section 5, we considered models trained with NFM that use noise injection levels $\sigma_{add} = 0.4$ and $\sigma_{mult} = 0.2$, whereas the ablation model uses $\sigma_{add} = 1.0$ and $\sigma_{mult} = 0.5$. Here, we want to better illustrate the trade-off between accuracy and robustness. We saw that there exists a potential sweet-spot where we are able to improve both the predictive accuracy and the robustness of the model. However, if the primary aim is to push the robustness of the model, then we need to sacrifice some amount of accuracy.

Fig. 10 is illustrating this trade-off for pre-actived ResNet-18s trained on CIFAR-10. We can see that increased levels of noise injections considerably improve the robustness, while the accuracy on clean data points drops. In practice, the amount of noise injection that the user chooses depend on the situation. If robustness is critical, than higher noise levels can be used. If adversarial examples are the main concern, than other training strategies such as adversarial training might be favorable. However, the advantage of NFM over adversarial training is that (a) we have a more favorable trade-off between robustness and accuracy in the small noise regime, and (b) NFM is computationally inexpensive, when compared to most adversarial training schemes. This is further illustrated in the next section.

### F.6 COMPARISON WITH ADVERSARIAL TRAINED MODELS

Here, we compare NFM to adversarial training in the small noise regime, i.e., the situation where models do not show a significant drop on the clean test set. Specifically, we consider the projected gradient decent (PGD) method (Madry et al., 2017) using 7 attack iterations and varying $l_2$ per-

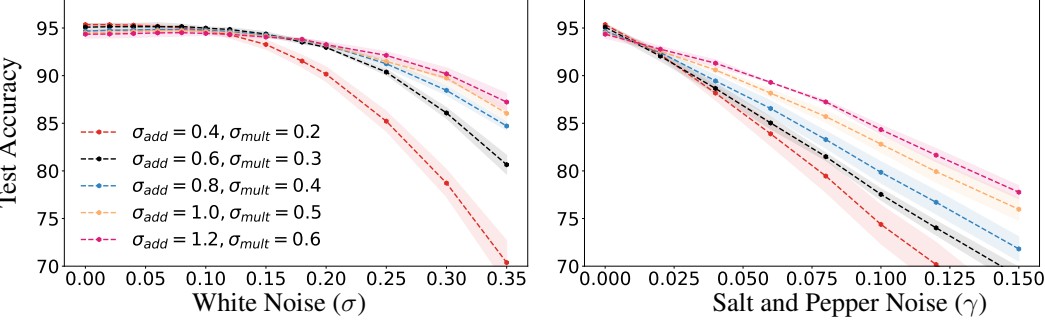

Figure 10: Pre-actived ResNet-18 evaluated on CIFAR-10 trained with NFM and varying levels of additive ($\sigma_{add}$) and multiplicative ($\sigma_{mult}$) noise injections. Shaded regions indicate one standard deviation about the mean. Averaged across 5 random seeds.

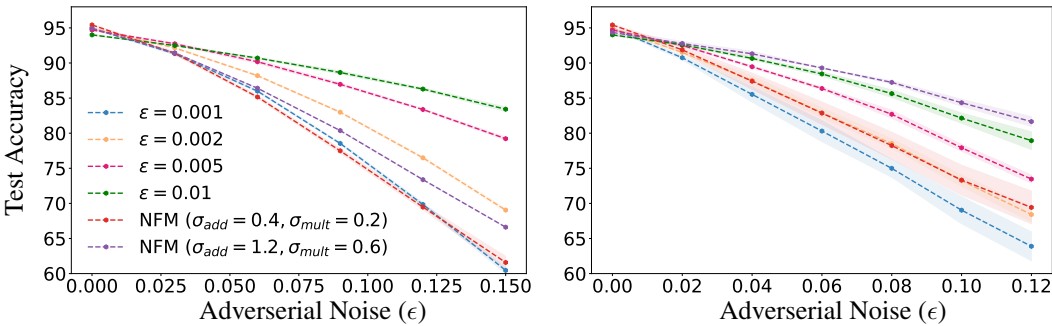

Figure 11: Pre-actived ResNet-18 evaluated on CIFAR-10 (left) and Wide ResNet-18 evaluated on CIFAR-100 (right) with respect to adversarial perturbed inputs. Shaded regions indicate one standard deviation about the mean. Averaged across 5 random seeds.

turbation levels $\epsilon$ to train adversarial robust models. First, we compare how resilient the different models are with respect to adversarial input perturbations during inference time (Fig. 11; left). Again the adversarial examples are constructed using the PGD method with 7 attack iterations. Not very surprisingly, the adversarial trained model with $\epsilon = 0.01$ features the best resilience while sacrificing about $0.5\%$ accuracy as compared to the baseline model (here not shown). In contrast, the models trained with NFM are less robust, while being about $1 - 1.5\%$ more accurate on clean data.

Next, we compare in (Fig. 11; right) the robustness with respect to salt and pepper perturbations, i.e., perturbations that both models have not seen before. Interestingly, here we see an advantage of the NFM scheme with high noise injection levels as compared to the adversarial trained models.

### F.7 FEATURE VISUALIZATION COMPARISON

In this subsection, we concern ourselves with comparing the features learned by three ResNet-50 models trained on Restricted Imagenet (Tsipras et al., 2018): without mixup, manifold mixup (Verma et al., 2019), and NFM. We can compare features by maximizing randomly chosen pre-logit activations of each model with respect to the input, as described by Engstrom et al. (2020). We do so for all models with Projected Gradient Ascent over 200 iterations, a step size of 16, and an $\ell_2$ norm constraint of 2,000. Both the models trained with manifold mixup and NFM use an $\alpha = 0.2$, and the NFM model uses in addition $\sigma_{add} = 2.4$ and $\sigma_{mult} = 1.2$. The result, as shown in Fig. 12, is that the features learned by the model trained with NFM are slightly stronger (i.e., different from random noise) than the clean model.

### F.8 TRAIN AND TEST ERROR FOR CIFAR-100

Figure 13 shows models trained with different training schemes on CIFAR-100. Compared to the baseline model, the models trained with manifold mixup and NFM have a similar convergence behavior. However, they are able to achieve a smaller test error. This shows that both manifold mixup and NFM have a favorable implicit regularization effect, where the effect is more pronounced for the NFM scheme.

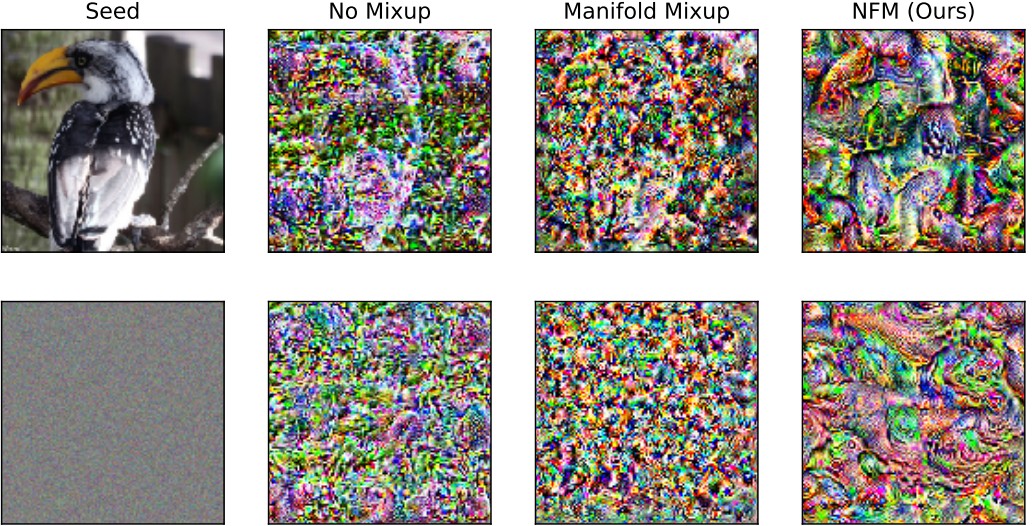

Figure 12: The features learned by the NFM classifier are slightly stronger (i.e., different from random noise) than the clean model. See Subsection F.7 for more details.

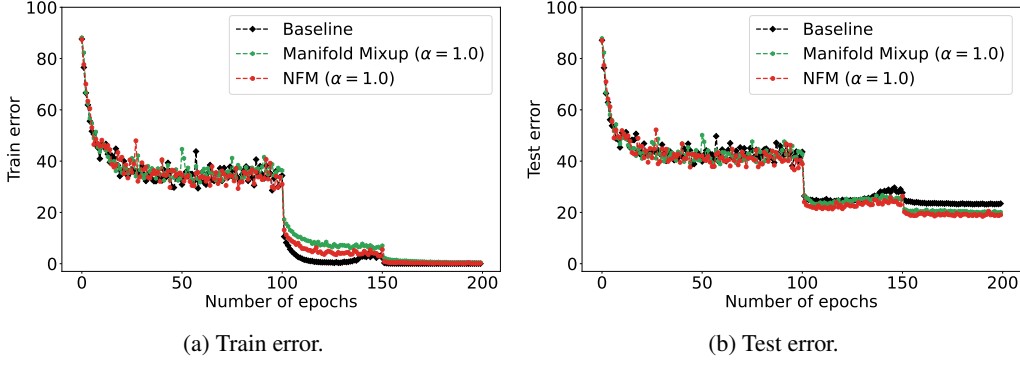

(a) Train error.                    (b) Test error.

Figure 13: Train (a) and test (b) error for a pre-actived Wide-ResNet-18 trained on CIFAR-100.

