# OpenReview forum: "Noisy Feature Mixup"
_ICLR.cc/2022/Conference — ICLR 2022 Poster_

### Official Review · Reviewer_gzuL · 2021-11-01

**Correctness:** 4
**Technical Novelty And Significance:** 3
**Empirical Novelty And Significance:** 3
**Recommendation:** 8
**Confidence:** 4

**Main Review:**

Overall this paper is well written and easy to follow. The strengths of this paper include:
1. This paper has a strong motivation, i.e., exploits the benefits of mixup and noise injection to further improve the generalization ability of a trained model.
2. The proposed method, i.e., Noisy Feature Mixup (NFM), is simple in principle and easy to implement in practice.
3. Theoretical analyses are offered to illustrate the underlying mechanisms of NFM. The theorems in the paper are built with reasonable assumptions, and the conclusions are sound and can explain the practical behaviors of NFM to some degree. Overall I find the theorems offer a good insight for NFM.
4. The proposed NFM is evaluated on different datasets, with two representative neural network architectures. The experimental designs are reasonable, and the results can well demonstrate the advantages of NFM over different Mixup variants.

For this paper, I have the following concerns/questions:
1. In NFM, the noisy mixup is performed on a random layer $k \in [L]$ in each iteration. But is it really necessary to apply NFM to every layer? In other words, is it possible that for some layers, the noisy mixup would lead to negative (or neutral) effects and hurt the performance?
2. From the experimental results, it is unclear if the stochasticity introduced by NFM during training would lead to a slow convergence of model training. Is it possible that such stochasticity would make the model difficult to converge, or require more time for the model to converge?
3. In Theorem 2, it is assumed that $E_{r \sim D_x}[g_k(r)] = 0$. However, I find this assumption could be too strong in practice.

**Summary Of The Paper:**

This paper proposes a new method for data augmentation, named Noisy Feature Mixup (NFM). This method combines the advantages of both the interpolation based training and noise injection schemes. In particular, this method is simple and easy to implement. Empirically, this paper shows that NFM achieves a favorable trade-off between the test accuracy on the clean set and the model robustness. Theoretically, it is shown that NFM enables smoother decision boundary, and amplifies the regularization effects of manifold mixup and noise injection. Additional theoretical analysis also shows that NFM training is approximately minimizing an upper bound related to the adversarial loss, thereby leading to more robust model.

**Summary Of The Review:**

The problem this paper focuses on is important, i.e., improve the generalization performance of deep models. To this end, this paper proposes a new method that exploits the benefits of mixup and noise injection, and is empirically demonstrated to be more effective at improving the accuracy and robustness of trained models. The proposed method is simple and easy to implement. Notably, it has sound theoretical analyses to illustrate the underlying mechanisms. Although for this paper there are still some points which are unclear, currently I think it is overall a good paper.

---

> ### Author Response · Authors · 2021-11-19
> **Thank you for your detailed review.**
>
> We would like to thank the reviewer for the careful review.
>
> > Is it really necessary to apply NFM to every layer? In other words, is it possible that for some layers, the noisy mixup would lead to negative (or neutral) effects and hurt the performance?
>
> That's an interesting question! In our experiments, we follow the setup of the authors of the manifold mixup paper for residual neural networks. It is not necessary to apply NFM to every layer. In fact, we only apply NFM to the outputs of the residual/attention blocks. We do not inject noise to the layers within the blocks. Further, it is possible that NFM leads to negative effects. For example, we saw a slight performance drop if we apply NFM to the final residual block's activations. Hence, given an architecture with N blocks, we consider only the first N-1 blocks for which we apply NFM. We did not further investigate other configurations.
>
> > Is it possible that such stochasticity would make the model difficult to converge, or require more time for the model to converge?
>
> In our experiments, models trained with NFM did not show slower convergence compared to the baseline model. Please see Figure 13 in the SM.
>
> > In Theorem 2, it is assumed that $E_{r~D_x}[g_k(r)]=0$. However, I find this assumption could be too strong in practice.
>
> This assumption could, in principle, be removed at the cost of resulting in a more complicated expression for the bound. We note that even with this assumption the presented bound is already quite complicated. The main purpose of Theorem 2 is to convey the idea that minimizing the NFM loss can lead to a small regularized adversarial loss, and it is best to achieve this by presenting compact results at the expense of generality in the main paper. For completeness, we have also included the result that one would have obtained when this assumption is removed (see Remark 1 in SM).

---

> > ### Comment · Reviewer_gzuL · 2021-11-29
> > **The authors have addressed my concerns**
> >
> > After reading the response from the authors, I think they have addressed my concerns.

---

### Official Review · Reviewer_o7KT · 2021-11-02

**Correctness:** 4
**Technical Novelty And Significance:** 3
**Empirical Novelty And Significance:** 2
**Recommendation:** 6
**Confidence:** 4

**Main Review:**

**Pros**
- The paper is easy to read and the writing is very clear.
- The paper provides thorough theoretical analyses, which the original manifold mixup paper lacks.
- The proposed method is effective in noisy test environments with various models and datasets.

**Cons**
- The performance gain under the clean setting is marginal.
- The experimental setting is limited. It will be valuable to evaluate methods in more general and natural corruption settings. (please refer https://github.com/hendrycks/robustness)
- Baselines in experiments are not sufficient. It will be informative to compare some recent mixup techniques enhancing robustness (e.g., Puzzle Mix [1] performs mixup-aware adversarial training, and AugMix [2] enhances corruption robustness).

**Additional comments**
- Does the theory can be generalized to the more general perturbations (such as natural data augmentation)?
- For me, the statement in Theorem 2 is counter-intuitive (L^NFM upper bounds the worst-case). It will be informative to add a more intuitive explanation for the theorem in the paper. (e.g., when the statement is valid).
- In appendix Figure 10, it is hard to interpret which one is better. If possible, the formal evaluation of the features will be also interesting.

[1] Kim et. al, Puzzle Mix: Exploiting Saliency and Local Statistics for Optimal Mixup, ICML 2020.
[2] Hendrycks et. al, AugMix: A Simple Data Processing Method to Improve Robustness and Uncertainty, ICLR 2020

**Post rebuttal**
I appreciate the authors' rebuttal and believe the revision improves the paper. However, the robustness experiments are still not convincing and require stronger baselines to be convincing (e.g., AugMix or other methods with stronger augmentations). Overall, I believe this is a clear paper and I maintain my score.

**Summary Of The Paper:**

This paper proposes noise added version of the manifold mixup. The authors verify the theoretical properties of the method in terms of regularization and robustness. The paper verifies the effectiveness of the proposed method on image classification tasks with noisy test environments.

**Summary Of The Review:**

This paper is well-written and provides thorough theoretical properties. However, the experiment settings are a bit limited and require more baselines to be compared.

---

> ### Author Response · Authors · 2021-11-19
> **Thank you for your careful review.**
>
> We especially appreciate the interesting comments that helped us to further improve and strengthen our paper.
>
> > The performance gain under the clean setting is marginal.
>
> Compared to the baseline model we improve the test performance by ~4% on CIFAR-100. The improvement compared to mixup or manifold mixup is less pronounced. However, we show that model robustness can be improved without additional computational overhead since the cost from noise injection was marginal.
>
> > It will be valuable to evaluate methods in more general and natural corruption settings.
>
> Great suggestion! We have run experiments on CIFAR10-C to demonstrate our method's robustness with respect to several perturbations and corruptions. Figure 6 demonstrates the advantage of our NFM scheme compared to other baselines on CIFAR-10-C.
>
> > It will be informative to compare some recent mixup techniques enhancing robustness.
>
> This is also an excellent suggestion that is aligned with the comment by Reviewer (WVNT). We provide additional results for CutMix and PuzzleMix on CIFAR-10, CIFAR-100, and CIFAR-10-C.
> Note that we trained models with CutMix and PuzzleMix with 200 epochs for a fair comparison; in contrast, the authors of the PuzzleMix paper used 1200 epochs for training.
>
> Both CutMix and PuzzleMix trained with 200 epochs achieve a higher test accuracy on CIFAR-10 than models trained with NFM, while being less robust than NFM. However, when training models with CutMix and PuzzleMix for 200 epochs on CIFAR-100, it can be seen that NFM achieves a better test accuracy on clean data.
>
> We also trained a PuzzleMix model for 1200 epochs on CIFAR-100, using the default parameters suggested by the PuzzleMix authors. We have successfully reproduced their results using their hyperparameters but note that this model is less robust to the perturbations that we considered.
>
> > Does the theory can be generalized to the more general perturbations (such as natural data augmentation)?
>
> Thanks for the interesting question. The theory is mainly based on Taylor expanding the loss function dependent on a small parameter that measures the magnitude of the perturbation. Therefore, we expect that the theory can be generalized to cover more general perturbations (of known mathematical form). We leave more thorough investigations into general perturbations to future work.
>
> > For me, the statement in Theorem 2 is counter-intuitive (L^NFM upper bounds the worst-case). It will be informative to add a more intuitive explanation for the theorem in the paper. (e.g., when the statement is valid).
>
> The statement is similar to the one made in Zhang et. al. (ICLR 2021). It conveys the key intuition that minimizing the NFM loss results in a small worst-case (adversarial) loss, allowing us to interpret its contribution to model robustness. We present some discussion on the theorem and its assumptions, highlighting this intuition; see Theorem 2. We have now added further discussion on the validity of the statement in SM (see the paragraphs after Theorem 4 in SM).
>
> > In appendix Figure 10, it is hard to interpret which one is better. If possible, the formal evaluation of the features will be also interesting.
>
> Doing a more formal evaluation of the features would be interesting indeed, but we feel that this is beyond the scope of this paper. Figure 10 was mainly intended as a teaser to visually showcase NFM's effects on the learned representation.

---

### Official Review · Reviewer_WVNT · 2021-11-02

**Correctness:** 3
**Technical Novelty And Significance:** 2
**Empirical Novelty And Significance:** 3
**Recommendation:** 8
**Confidence:** 3

**Details Of Ethics Concerns:**

None.

**Main Review:**

## Strengths:

- The main contribution of this paper is extending input mixup (Zhang et al. (ICLR'18)) and manifold mixup (Varma et al. (ICML'19)) to Noisy Feature Mixup, which applies mixup randomly to *all layers of the neural network* and *adding random noise to the mixup examples*.
- Through this extension, the authors find that Noisy Feature Mixup improves the robustness to several kinds of noise models compared to previous mixup variants.
- Additionally, a theoretical analysis of the regularization effects of Noisy Feature Mixup is presented, building on top of the results by Zhang et al. (ICLR'21).

**Related works**

The related works are comprehensive in this area.

## Weakness:

The main weakness of this paper is the accessibility of the technical contribution. Thus, I think several revisions towards mitigating this weakness could improve the presentation, including, for example

- Illustrative examples and figures to convey Theorem 1 and Theorem 2, for example, similar to Figure 2 in Zhang et al. (ICLR'2021).
- The proofs are in line with recent works including Zhang et al. (ICLR'2021) which seem correct to me. The main idea is using the Taylor's expansion to the Noisy Feature Mixup objective. Then, the authors note that the random noise injection can be simplified on top of the regularization terms from Zhang et al. (ICLR'2021). However, the steps are difficult to verify; in particular, more detailed derivations should be included in the appendix.
    - For example, above equation (30), it's stated that "we compute, using linearity and chain rule." However, it's not clear to me how this is computed.
    - Then after equation (33), it's stated that "The equation in the theorem follows upon setting..." It's not immediately clear to me how this step is carried out either.
    - The rest of the appendix has similar issues as above. They should be expanded with more clear explanations and derivations.

**Missing ablation studies**

- It's unclear why the authors set the hyper parameters of the additive noise $\sigma_{add}=0.4$ and the multiplicative noise $\sigma_{mult}=0.2$. This should be justified with an ablation study that varies each hyper parameter.
- Since there are two components in the proposed approach (all-layer mixup and random noise injection), It's also unclear which part is contributing more or less. Again there should be an ablation study to justify the role of each component in the proposed approach.

**Missing baselines**

- The authors mainly consider Mixup and Manifold Mixup as baselines in the experiments. This is marginally acceptable. To make the experiments stronger, I think the authors could consider other related methods for improving robustness such as *label smoothing*.

**Summary Of The Paper:**

This paper studies data augmentation methods for improving the robustness of supervised learning. The main contribution is presenting Noisy Feature Mixup, extending input mixup and manifold mixup to all layers of a neural net. The experimental results show that the proposed approach improves the robustness of supervised learning under several noise attacks on the input data set. The theoretical results derive the Taylor's expansion of the Noisy Feature Mixup optimization objective.

**Summary Of The Review:**

This is a technically solid and empirically strong paper. The theoretical results follow from recent works for mixup (Zhang et al. (ICLR'21)). The experimental results are conducted on three popular image datasets, with three input perturbation models, and do support the validity of the proposed approach. The main weakness of this paper is the accessibility of the technical contribution and the lack of ablation studies. I think several revisions could help mitigate these weakness.

---

> ### Author Response · Authors · 2021-11-19
> **We are very thankful for the high appraisal of our work.**
>
> We would like to thank the reviewer for the time and effort spent reviewing this paper.
>
> > Illustrative examples and figures to convey Theorem 1 and Theorem 2, for example, similar to Figure 2 in Zhang et al. (ICLR 2021).
>
> We thank the reviewer for this suggestion. We have now included a plot similar to Figure 2 in Zhang el. Al. (ICLR 2021) (see Figure 7 in SM) to show that the second order Taylor approximation of the NFM loss is generally close to the original NFM loss during both training and testing.
>
> > "we compute, using linearity and chain rule." However, it's not clear to me how this is computed.
>
> We have now included all the details of the computation in the proof, which should help readers not acquainted with Zhang et. al. (ICLR 2021) to better follow our derivations.
>
> > "The equation in the theorem follows upon setting..." It's not immediately clear to me how this step is carried out either.
>
> We have now included more details and discussed intermediate steps of our derivations.
>
> > It's unclear why the authors set the hyper parameters of the additive noise $\sigma_{add}=0.4$ and the multiplicative noise $\sigma_{mult}=0.2$.
>
> Figure 10 in Appendix F.6 shows how the performance of models trained with NFM varies as a function of different noise levels. Increasing the noise levels improves the robustness, while the accuracy on clean data slightly drops.  In our experiments, we decided to use the parameters 0.4/0.2, since those provide a good trade-off between performance on clean data and increased robustness. Our aim was to show that a model can achieve state-of-the-art performance on clean data (or at least improve accuracy compared to the baseline model), while being more robust. In addition, we show a model (indicated by $*$) that was trained with noise levels 1.0/0.5. Further, we do not fine-tune the noise levels to a specific problem or architecture to show that the parameters are insensitive to the specific task / model.
>
> > This should be justified with an ablation study that varies each hyper parameter.
>
> Agreed. We now provide an ablation study in Table 5 of Appendix F 5. This table along with Figure 10 helps to disentangle the different components and illustrate the benefits of our Noisy Feature Mixup scheme. Notably, using only noise injections or just manifold mixup is less favorable than their combination.
>
>
> > The authors mainly consider Mixup and Manifold Mixup as baselines in the experiments. This is marginally acceptable. To make the experiments stronger, I think the authors could consider other related methods for improving robustness such as label smoothing.
>
> This is a valid point, and also relates to a comment by Reviewer (o7KT). With the addition of label smoothing, we provide comparisons to CutMix and PuzzleMix on CIFAR-10.
>
> We evaluate label smoothing parameters in [0.05, 0.1, 0.2]. The model trained with 0.1 showed the best trade-off; as demonstrated, label smoothing is slightly improving the generalization performance and robustness of the model, but the gains are not significant. Due to the weak performance of label smoothing, and limited space, we do not show label smoothing results for CIFAR-100. Instead, we compare NFM to CutMix and PuzzleMix, and our results indicate that the performance of NFM is favorable. Please also find additional results for CIFAR-10-C.

---

> > ### Comment · Reviewer_WVNT · 2021-11-29
> > **Thanks for the revision**
> >
> > I'm glad to see that the revision has taken into account my concerns above. Figure 7 in particular gives some confidence to the correctness of your result without having to fully verify the proof; thus, I retain my generally positive assessment of your work. Please find a few minor comments below:
> > - The citation for Zhang et al. (2020) "How does mixup help with robustness and generalization?" in Section B.1 should be Zhang et al. (2021) instead.
> > - In Figure 7, the loss curve for NFM has some bumps due to randomness but the loss curve with the approximated loss is flat. Could you add an explanation about how you plot the approximated loss? Was this because you have taken the expectation over the mixing weight in the approximated loss?

---

> > > ### Author Response · Authors · 2021-11-29
> > > **Thank you very much for your positive assessment**
> > >
> > > Thank you for pointing out the typo in the citation. As for Figure 7, you are right. We have taken the expectation over the mixing weights in the approximated loss.

---

### Official Review · Reviewer_46ws · 2021-11-05

**Correctness:** 3
**Technical Novelty And Significance:** 3
**Empirical Novelty And Significance:** 3
**Recommendation:** 6
**Confidence:** 3

**Main Review:**

Theorem 1 is one of the most important contributions of this paper. It shows that minimizing the proposed NFM loss is equivalent to minimizing the sum of the original loss and feature-dependent regularizers. The form of the regularizers is very complicated. I have checked part of them and seems to be correct. Too much work to follow the rest part. This theorem shows that the regularizers actually reduce the Jacobians and Hessians, which is reasonable and fit everyone's perception. In my opinion, although there is no surprising conclusion, this contribution provides evidence for everyone’s understanding of the mixup type methods.

The experiment part provides the necessary experiments to support the argument. But I am curious about the different behaviours of the proposed methods for features of different levels. According to conjecture, when the feature level is different, the meaning of adding noise and mixup may become completely different. It seems that this part of the discussion is missing.

**Summary Of The Paper:**

The paper proposes a new and inexpensive mixup method named Noisy Feature Mixup (NFM) to mitigate over-fitting and improve generalization. NFM combines mixup and noise injection, which inherits the benefits of these methods. Due to its conciseness, it is convenient to apply NFM in model training. More importantly, the authors prove NFM's regularization effect on model optimization and robustness improvement with mathematical derivation. They build the implicit connection between the NFM empirical loss and the original loss. Then they identify the regularizing effects of the proposed NFM. To demonstrate that NFM helps robustness, the authors relate the NFM loss to the one used for adversarial training, which can be regarded as an example belonging to distributionally robust optimization. With integrated experiments, NFM shows its superiority on various models and datasets. The further discussion shows the interesting tradeoff between predictive accuracy on clean and perturbed test sets. The Supplementary Material provides detailed proof and additional experimental results to show NFM's firmed ground.

**Summary Of The Review:**

It is difficult to rate this paper. Theorem 1 has theoretical value. The proposed method is relatively simple (both pros and cons). But I would vote positively for theorem 1.

---

> ### Author Response · Authors · 2021-11-19
> **We would like to thank the reviewer for the time and effort writing this review.**
>
> > Theoretical value.
>
> Thank you very much for appreciating the value of Theorem 1. We have now included a lot more details in the proof of Theorem 1, hoping that the derivation is easier to follow for readers not acquainted with Zhang et. al. (ICLR 2021). We would also like to emphasize that, in addition to Theorem 1, Theorem 2 is equally valuable for understanding how NFM can lead to model robustness, which is one of our most important contributions in the paper. We hope that the reviewer can recognize the importance of Theorem 2 and its implications (both theoretically and empirically).
>
> > I am curious about the different behaviours of the proposed methods for features of different levels. According to conjecture, when the feature level is different, the meaning of adding noise and mixup may become completely different. It seems that this part of the discussion is missing.
>
> The effects of NFM indeed vary with the features used, just like how the effects of manifold mixup vary with the features (hidden representations of the layers of neural networks). The important question/discussion here is not so much about how different these effects/meanings are (quantifying them would be very challenging, depending on the network architectures), but rather on how these effects can improve the hidden representations and decision boundaries of neural networks at multiple layers. We focus on studying the latter via the lens of implicit regularization and distributionally robust optimization in the paper.

---

### Author Response · Authors · 2021-11-19
**General Response**

We would like to thank all reviewers for the positive and constructive feedback. The reviews have helped us to revise and strengthen the paper. In summary, we now:

- improve the presentation of our proofs by providing more details and showing intermediate steps;
- provide results for additional baselines, including CutMix and PuzzleMix;
- provide results for CIFAR-10-C, showing that Noisy Feature Mixup (NFM) improves model robustness with respect to a wide range of perturbations and corruptions;
- provide an ablation study that demonstrates the favorable interaction of mixing features and noise injections.

We provide a detailed response to all comments below.

---

> ### Comment · Reviewer_o7KT · 2021-11-25
> **There is no comparison with AugMix**
>
> I appreciate the authors' additional experiments on robustness. However, the current updated version seems not to contain AugMix results (In the second bullet, authors mentioned they included AugMix results).

---

> > ### Author Response · Authors · 2021-11-25
> > **PuzzleMix**
> >
> > Thanks for pointing this out. We meant CutMix and PuzzleMix.

---

### Decision · Program_Chairs · 2022-01-20

**Decision:**

Accept (Poster)

**Comment:**

This paper introduces Noisy Feature Mixup: an extension of input mixup and manifold mixup to all layers of a neural net, for the purpose of improving robustness and generalization in supervised learning. Experimental validation supports the increased robustness to attacks on the input data. The reviewers find the paper well written and they appreciate the theoretical analysis as well as the empirical results. The reviewers did not identify any big problems, and their minor concerns were sufficiently addressed in the author reponse. I'm therefore happy to recommend accepting this paper.